# Increased hydraulic risk in assemblages of woody plant species predicts spatial patterns of drought-induced mortality

Pablo Sanchez-Martinez [1,2,3] ✉, Maurizio Mencuccini [2,4], Raúl García-Valdés [2,5], William M. Hammond[6], Josep M. Serra-Diaz [7,8,9], Wen-Yong Guo[10,11], Ricardo A. Segovia [12,13], Kyle G. Dexter [3,14], Jens-Christian Svenning [11], Craig Allen [15] & Jordi Martínez-Vilalta [1,2]

Predicting drought-induced mortality (DIM) of woody plants remains a key research challenge under climate change. Here, we integrate information on the edaphoclimatic niches, phylogeny and hydraulic traits of species to model the hydraulic risk of woody plants globally. We combine these models with species distribution records to estimate the hydraulic risk faced by local woody plant species assemblages. Thus, we produce global maps of hydraulic risk and test for its relationship with observed DIM. Our results show that local assemblages modelled as having higher hydraulic risk present a higher probability of DIM. Metrics characterizing this hydraulic risk improve DIM predictions globally, relative to models accounting only for edaphoclimatic predictors or broad functional groupings. The methodology we present here allows mapping of functional trait distributions and elucidation of global macro-evolutionary and biogeographical patterns, improving our ability to predict potential global change impacts on vegetation.

A substantial number of woody plant assemblages worldwide are experiencing increased mortality due to rising drought severity and temperature (termed drought-induced mortality, DIM), driven by anthropogenic climate change[1–3]. Such mortality modifies ecosystem composition, structure and functioning[4], with large impacts on biodiversity and biogeochemical cycles[5,6]. Generally, DIM is triggered by hydraulic failure[7–10], a physiological process causing loss of functionality of the plant conductive tissue (xylem), eventually leading to desiccation and death. Previous studies have shown that plant hydraulic traits have the potential to improve our capacity to understand and predict DIM[11] and drought impacts on ecosystem fluxes[12,13], as well as the community dynamics[14,15] emerging from these processes. Accordingly, hydraulic schemes are being incorporated into forest vulnerability assessments[16,17] and vegetation models, from the regional[18,19]

[1]Universitat Autònoma de Barcelona, Cerdanyola del Valles, Barcelona, Spain. [2]CREAF, Cerdanyola del Valles, Barcelona, Spain. [3]School of GeoSciences, University of Edinburgh, Edinburgh, UK. [4]ICREA, Barcelona, Spain. [5]Department of Biology and Geology, Physics and Inorganic Chemistry, Rey Juan Carlos University, Móstoles, Madrid, Spain. [6]Agronomy Department, University of Florida, Gainesville, FL, USA. [7]Université de Lorraine, AgroParisTech, INRAE, Nancy, France. [8]Eversource Energy Center, University of Connecticut, Storrs, CT, USA. [9]Department of Ecology and Evolutionary Biology, University of Connecticut, Storrs, CT, USA. [10]Research Center for Global Change and Complex Ecosystems & Zhejiang Tiantong Forest Ecosystem National Observation and Research Station, School of Ecological and Environmental Sciences, East China Normal University, Shanghai, P. R. China. [11]Department of Biology, Center for Ecological Dynamics in a Novel Biosphere (ECONOVO) & Center for Biodiversity Dynamics in a Changing World (BIOCHANGE), Aarhus University, Aarhus C, Denmark. [12]Institute of Ecology and Biodiversity (IEB), Santiago, Chile. [13]Departamento de Botánica, Facultad de Ciencias Naturales y Oceanográficas, Universidad de Concepción, Concepción, Chile. [14]Royal Botanic Garden Edinburgh, Edinburgh, UK. [15]Department of Geography and Environmental Studies, University of New Mexico, Albuquerque, NM, USA. ✉e-mail: p.sanchez@creaf.uab.cat

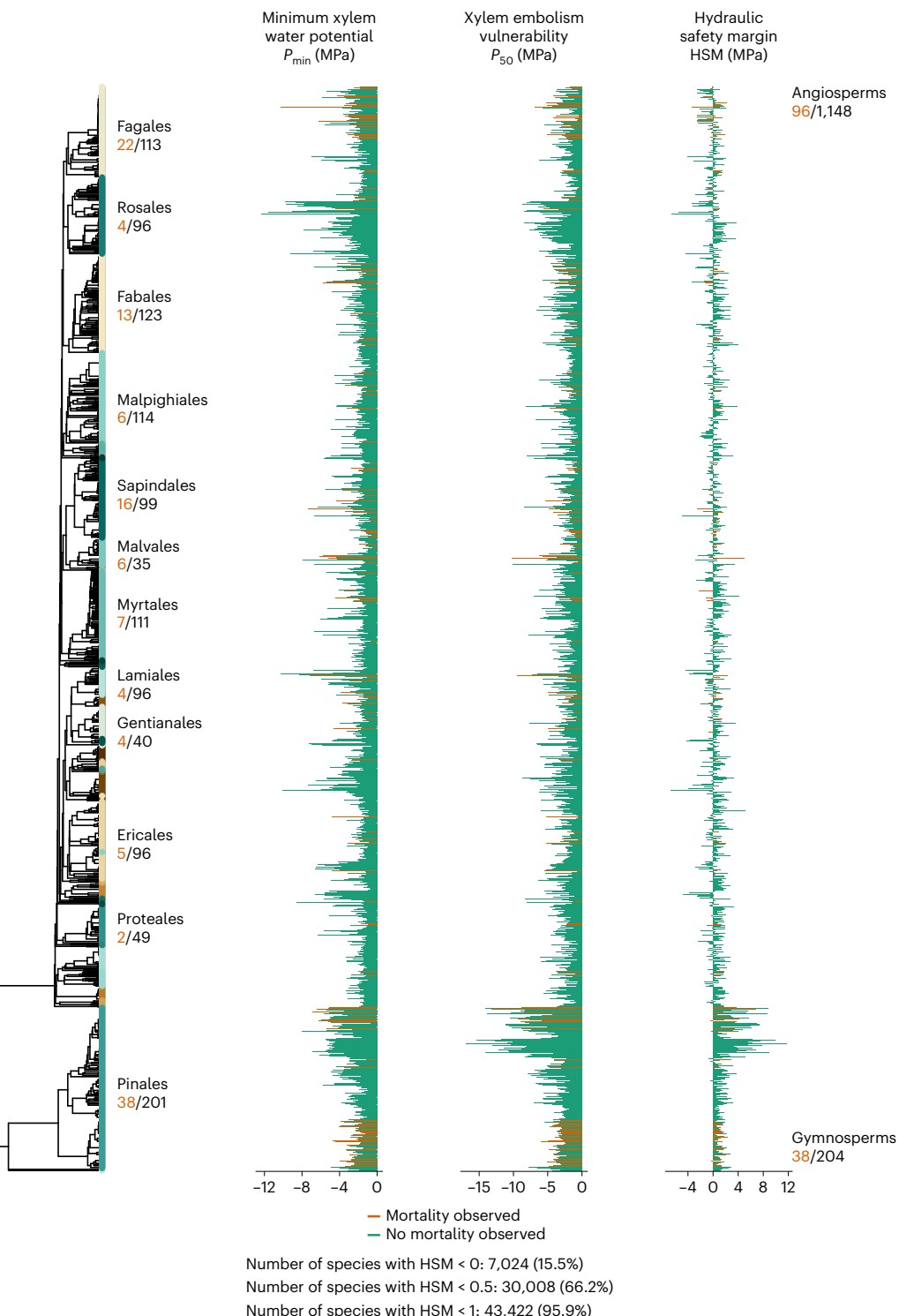

**Fig. 1 | Phylogenetic distribution of imputed hydraulic traits for species with observed xylem minimum water potential and/or xylem embolism vulnerability.** Dark orange, species with observed mortality. Green, species without observed mortality. The most important order names are shown. The total number of species with trait data is shown in black and the number in dark orange is the number of those species that have an observed mortality event.

to the global[20] scale. However, the predictive capacity of these models is still poor[18,21,22], potentially reflecting lack of high-quality hydraulic data or insufficient understanding of the mechanisms involved.

Hydraulic dysfunction happens when drought stress exceeds the capability of the xylem to tolerate high tensions (low water potentials), generating emboli in conduit lumens that disrupt water flow. This disruption can lead to hydraulic failure if embolism propagates[23]. The

probability of suffering hydraulic failure (that is, hydraulic risk)[11,24] is commonly quantified with the hydraulic safety margin (HSM), which is the difference between the minimum observed water potential in the xylem ($P_{min}$, a measure of drought exposure reflecting plant hydraulic regulation at the tissue level) and the water potential causing 50% or 88% of hydraulic conductivity loss ($P_{50}$ and $P_{88}$, measuring vulnerability to xylem embolism)[25,26]. HSM is thus an individual- and site-specific

physiological metric likely to be associated with DIM. However, data availability of $P_{min}$, $P_{50}$ and $P_{88}$ at broad spatial scales is scarce both across, and especially, within species, and frequently available data do not reflect local conditions. Not surprisingly, the species HSM is generally a poor predictor of their mortality and only improves marginally existing models[18,21].

The distribution of HSM values within woody plant assemblages has been shown to relate to their response to extreme drought events[12,27,28] and to the maintenance of productivity under increasing drought[15]. This functional variability is probably explained by the variety of existing species-specific mechanisms to cope with drought[29], influenced in turn by environmental filtering and evolutionary legacies present in any species assemblage[30]. Here, we posit that our capacity to predict mortality occurrence will be improved by considering the variability of hydraulic risk at the site level (assemblages of potentially co-occurring species) and not only the average hydraulic risk of individual species in the assemblage. However, $P_{50}$ and $P_{min}$ data are only available for 1,678 and 819 woody plant species, respectively, representing less than 1.5% of the world's estimated number of woody plant species. Nonetheless, we have recently shown that $P_{min}$ and $P_{50}$ are phylogenetically conserved to a substantial degree and are related to edaphoclimatic affiliations[31]. Including phylogenetic and edaphoclimatic information is therefore likely to improve the trait imputations required to provide global trait coverage. These results, together with increased availability of plant distribution data, pave the way towards predictions of hydraulic risk metrics that cope with the data scarcity problem, allowing to move from individual species predictions to analyses of species assemblages at the global scale.

Here, we use a new global database of hydraulic traits[32] and edaphoclimatic and phylogenetic information coupled with random-forest modelling[33] to estimate drought exposure ($P_{min}$) and xylem drought resistance ($P_{50}$ and $P_{88}$) and hence hydraulic risk (HSM), for 44,901 woody plant species. We georeferenced these predictions using species distribution data[34] and mapped aggregated hydraulic metrics for species assemblages at a 5 km resolution, globally. Then, we used linear models to test which metrics of hydraulic risk characterization (species-assemblage mean and minimum hydraulic risk, its variability and the number of species with high hydraulic risk) can predict observed DIM, using precisely georeferenced records of DIM occurrence[3]. Finally, we use maximum entropy models[35] to project DIM occurrence probability worldwide using different edaphoclimatic predictors and the newly derived hydraulic metrics. We propose that species-assemblage hydraulic risk metrics will predict DIM occurrence, reflecting both that species with lower HSM incur greater mortality risk and that assemblages with a higher number of species at hydraulic risk will experience more DIM. By applying this framework, we provide a global projection of woody plant hydraulic risk and associated DIM.

## Results and discussion
### Widespread low HSMs in woody plants
Random-forest models[33], considering phylogenetic data jointly with edaphoclimatic affiliations and trait covariation, had substantial predictive power for species-specific minimum xylem water potential ($P_{min}$) and vulnerability to embolism ($P_{50}$) with a cross-validation $R^2$ of $0.60 \pm 0.10$ and $0.54 \pm 0.12$, respectively (mean and standard deviation; Supplementary Table 1; Methods). Estimated species HSM was related to observed HSM values, with an $R^2$ of 0.51. Overall, 7,024 out of 44,901 species (15.5%) presented negative HSM values, 66.2% of all species had HSM < 0.5 MPa and 95.9% of all species had HSM < 1 MPa (Fig. 1 and Supplementary Fig. 1). These results generalize previous studies[25] indicating convergence towards low mean HSM in woody plants, pointing to a prevalent strategy of maximizing the usage of available water, fixing carbon at the expense of increasing hydraulic risk. Negative HSM implies embolism levels above 50%, which are expected to be stressful, especially for gymnosperms[36]. Some species

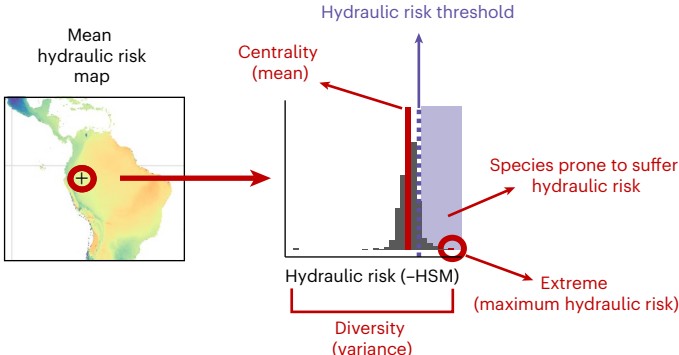

**Fig. 2 | Example of species assembly hydraulic risk composition.** Schematic representation of species assemblies data, from which the hydraulic metrics are extracted and mapped.

(particularly angiosperms) may be adapted to recover from embolism by refilling conduits, resprouting from branch nodes below dead tissues or radial growth following drought relief[11]. When using $P_{88}$ instead of $P_{50}$ for angiosperms, which may be a more realistic hydraulic failure threshold for angiosperm species ($P_{50/88}$ results hereafter)[36], only 165 species out of 44,901 species (0.37%) presented negative $HSM_{50/88}$ values (HSM calculated using $P_{50}$ for gymnosperms and $P_{88}$ for angiosperms; Supplementary Fig. 2).

### Species hydraulic risk is a poor predictor of mortality
We did not find significant relationships ($P > 0.3$) between species hydraulic safety margin (HSM or $HSM_{50/88}$) and species DIM. This result supports the lack of a strong relationship at broad spatial scales between species' mean-hydraulic risk and their mortality[18]. However, we found significant negative relationships of species HSM (slope = −0.16, s.e. = 0.03, $P < 0.001$) and $HSM_{50/88}$ (slope = −0.34, s.e. = 0.02, $P < 0.001$) with the number of recorded DIM events per species. These relationships were significant for both angiosperms and gymnosperms, even though their predictive power was low (pseudo-$R^2 < 0.15$ and area under the curve (AUC) < 0.57 in both cases). Equivalent results were obtained when using only observed HSM values (that is, excluding imputed values). These results together suggest that, even though species with low HSM tend to present a higher number of recorded DIM events, this information is not sufficient to predict with reasonable accuracy the DIM of species. This may be because not only mean species hydraulic risk but also local environmental conditions are playing a crucial role in determining mortality risk. Thus, incorporating a geographical perspective may improve predictive capacity of DIM occurrence.

### Characterizing species assemblages hydraulic risk
We aggregated observed and imputed data for species xylem minimum water potential ($P_{min}$) and embolism vulnerability ($P_{50}$ and $P_{88}$) into species assemblages expected by species distribution data in $5 \times 5$ km² grid cells (Fig. 2 and Supplementary Fig. 3a,b)[34] (Methods). Areas with high drought incidence such as the Mediterranean basin, southwestern Africa, southwestern United States and southwestern Australia presented species assemblages with lower vulnerability to embolism (lower mean $P_{50}$) (Fig. 3a) but not necessarily lower hydraulic risk (constant mean HSM) (Fig. 4a; note that hydraulic risk is represented as negative HSM so higher values represent higher risk). This pattern underlines the importance of tissue-level drought exposure ($P_{min}$, Supplementary Fig. 4) in determining hydraulic risk, as species can converge towards similar HSM even when being exposed to very different levels of climatic drought or present very different HSM under the same conditions depending on their functional strategies[37]. However, species presenting the highest hydraulic risk were found in places with high drought

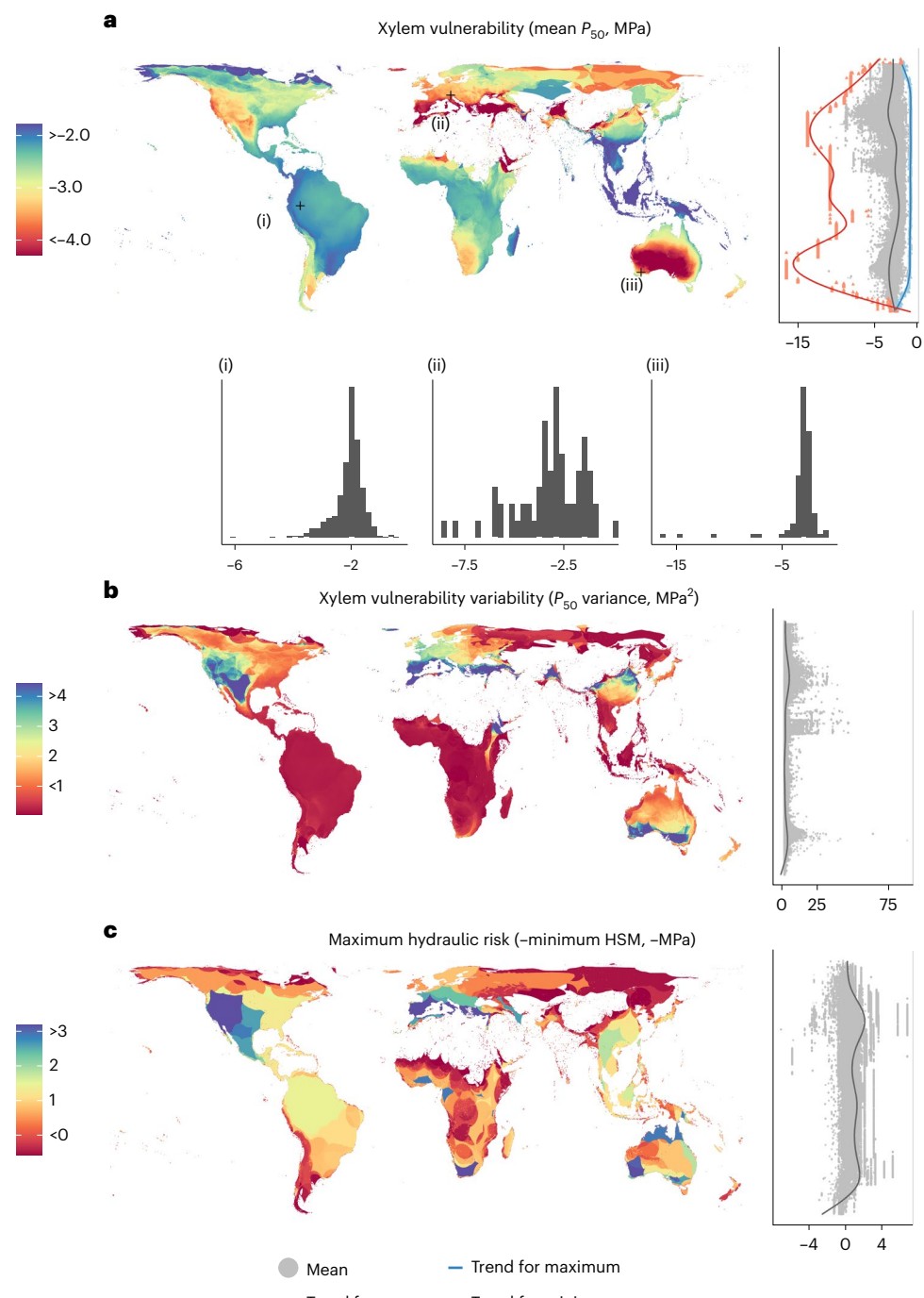

**Fig. 3 | Global distribution of species-assemblage hydraulic metrics and their latitudinal patterns. a**, Mean xylem vulnerability ($P_{50}$). **b**, $P_{50}$ variance. **c**, Maximum hydraulic risk represented as negative minimum HSM. The distribution of species-level values from which metrics are calculated for a sample of three representative pixels are shown in histograms in **a**. Lateral scatterplots show the distribution of pixel values. Trend lines for pixel values are shown for scatterplots by means of generalized additive model (GAM).

incidence (for example, the Mediterranean basin, western United States, Mexico, southwestern Australia and southern Africa; Fig. 3c), probably a result of the combination of high exposure and occurrence of some sensitive species at those locations. The apparent invariance of maximum hydraulic risk over some large areas (for example, the Amazon basin; Fig. 3c) probably results from species with particularly low HSM values having widespread distributions. In some cases, these results may be influenced by limited data availability together with relatively low species diversity (for example, boreal forests in Russia).

Maintaining a reasonably high HSM may imply very different strategies, including high embolism resistance but also deep roots, tight stomatal regulation or drought deciduousness to limit $P_{min}$. The implications of these strategies may not be equivalent, which is a matter that requires further study. For example, in the case of stomatal and leaf area regulation, the carbon balance is also impacted directly, which could potentially result in indirect effects on the hydraulic system that could promote dehydration in the longer term or carbon starvation[10,38,39]. While hydraulic failure has been ubiquitously

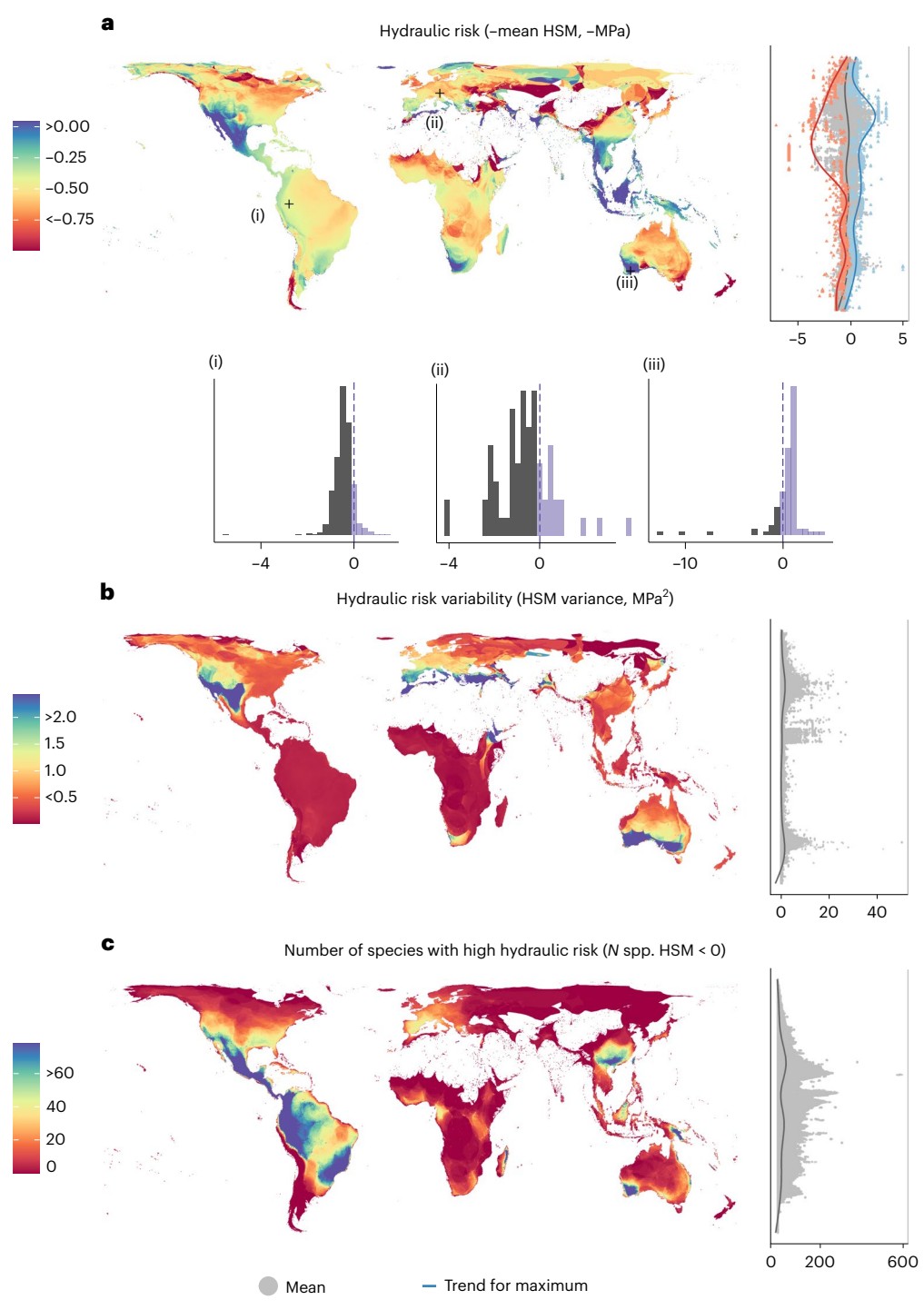

**Fig. 4 | Global distribution of species-assemblage hydraulic metrics and their latitudinal patterns. a**, Mean-hydraulic risk represented as −HSM. **b**, HSM variance. **c**, Number of species with negative HSM values. The distribution of species-level values from which metrics are calculated for a sample of three representative pixels are shown in histograms in **a**. Lateral scatterplots show the distribution of pixel values. Trend lines for pixel values are shown for scatterplots by means of GAM.

associated with drought-induced tree mortality[8,9,40], a high proportion of studies on DIM have also shown substantial reductions in total plant non-structural carbon, that is, a potential signal for carbon starvation[38]. At present, there is not a clear species-level or coarser-scale threshold for this mechanism of tree mortality, leaving it out of reach for trait-based models of DIM. However, including drought length and intensity in future studies might be useful to deepen our understanding of the consequences of changing drought intensities, which are most likely to invoke stronger interactions between carbon limitations and hydraulics or in extreme cases may result in greater mortality risk for plants from carbon starvation.

The functional diversity of species assemblages was further characterized by estimating the variability of strategies in a community (trait variance at the grid cell level). The highest variability for both

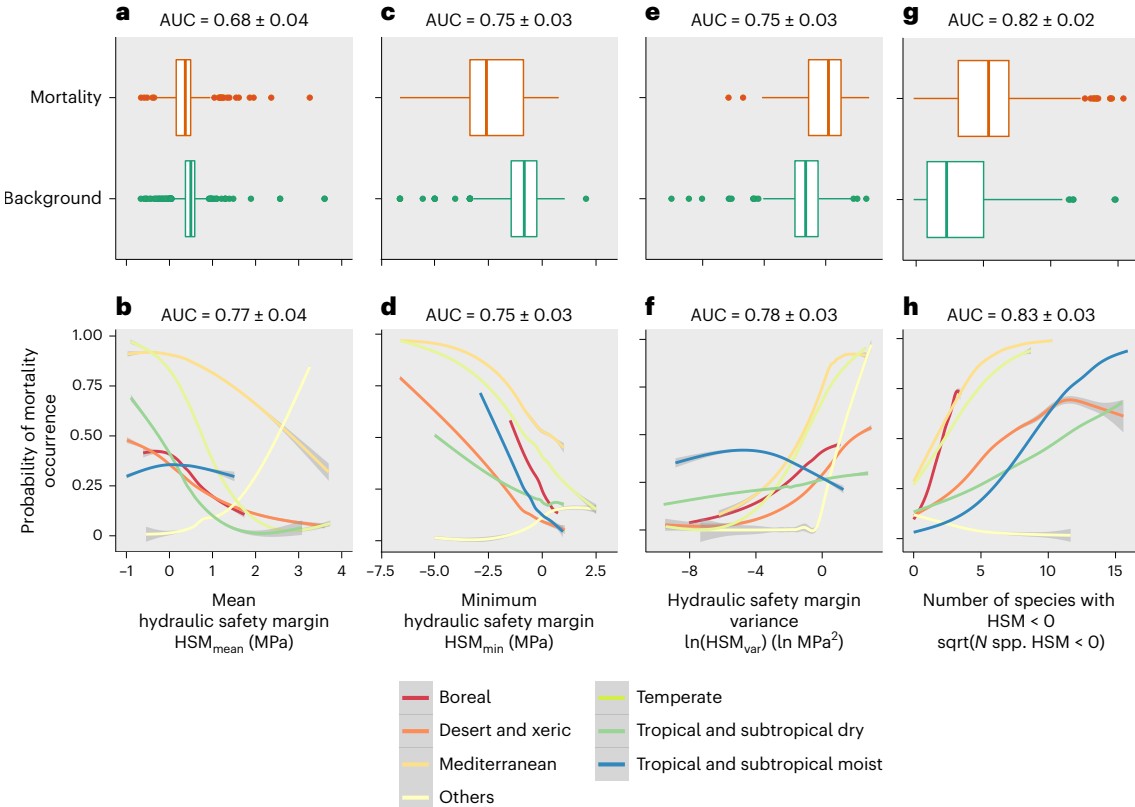

**Fig. 5 | Relationships between DIM occurrence and species-assemblage hydraulic metrics. a–h,** Represented are: mean HSM (**a,b**), minimum HSM (**c,d**), HSM variance (**e,f**) and the number of species with HSM < 0 (**g,h**) excluding (**a,c,e,g**) and including (**b,d,f,h**) their interaction with biome. Results summarize 100 iterations of generalized linear models in each case. In each iteration, a different random set of background points was sampled. **a,c,e,g,** Boxplots show species-assemblage metrics values for pixels with mortality compared to background locations. Boxplots represent first, second and third quartiles and whiskers represent maxima and minima. **b,d,f,h,** Mean response curves and the 95% coefficient interval for species-assemblage metrics for each biome. sqrt, square root.

$P_{50}$ and HSM was found in grid cells with relatively high drought incidence (for example, the Mediterranean basin, western United States, northern Mexico, southern Australia, Turkey and the Yemen in Figs. 3b and 4b), generalizing previous findings at regional scales[41–44]. We observed a spatial decoupling at the global scale between hydraulic trait variability and species richness. While species richness peaks in highly favourable habitats without water limitations[45] (Supplementary Fig. 3), hydraulic trait variability is higher where water scarcity leads to different physiological solutions to cope with drought in different plant lineages, resulting in a wide range of hydraulic trait values[42,43]. These results are in contrast with the favourability proposal[46] and previous results showing a higher functional diversity towards the equator in some traits[47] but are aligned with other results showing that evolutionary, and potentially functional, diversity peaks under intermediate precipitation[48,49]. Functional diversity may increase in sites with some degree of resource limitation which in turn allows the coexistence of lineages presenting different drought-coping strategies (for example, the case of the coexistence of gymnosperms such as *Pinus* spp. and angiosperms such as *Quercus* spp. in Mediterranean forests, with their divergent hydraulic strategies)[42,50]. However, this particular result may be influenced by higher sampling in areas with more severe droughts and needs to be confirmed by further studies.

We further characterized the hydraulic risk of species assemblages by calculating the number of species presenting HSM < 0, as another species-assemblage-specific hydraulic risk threshold. This metric represents the number of species expected to experience hydraulic dysfunction, potentially providing meaningful information on the likelihood of a site experiencing DIM. The number of species presenting HSM < 0 at the pixel level was highly variable (Fig. 4c), showing potential to characterize hydraulic risk at the species-assemblage level. Projections showed that species assemblages with a high number of species with HSM < 0 occur both in dry and wet places (for example, Mexico and western Amazonia, respectively).

Results based on $HSM_{50/88}$ projections were similar but showed a lower total number of species with negative values. These results showed lower HSMs in boreal forests, which may be due to the dominance of gymnosperms in this biome and that $P_{50}$ (the value used for gymnosperms for $HSM_{50/88}$) may be easier to surpass compared to $P_{88}$ (the value used for angiosperms) (Supplementary Fig. 5).

**Species-assemblage hydraulic risk is related to mortality**

We found significant relationships ($P < 0.01$) between species-assemblage hydraulic risk metrics and DIM (Fig. 5). Compared to species HSM, species-assemblage hydraulic risk metrics had higher predictive power for DIM occurrence (pseudo-$R^2$ between 0.07 and 0.47, AUC between 0.68 and 0.84) and far outperformed the predictive power of a climatic aridity index, annual precipitation and maximum temperature (pseudo-$R^2 < 0.02$, AUC < 0.6) (Supplementary Table 2). The relationships of hydraulic metrics with DIM for species assemblages remained significant even after the climatic aridity index was included in the models as a covariate (Supplementary Table 3). These results indicate that metrics related to the hydraulic risk of local species assemblages incorporate meaningful information beyond the local drought status. The relationships between DIM occurrence and hydraulic risk metrics of species assemblage were highly consistent across different biomes and plant functional types (PFTs) (Fig. 5 and Supplementary Fig. 6).

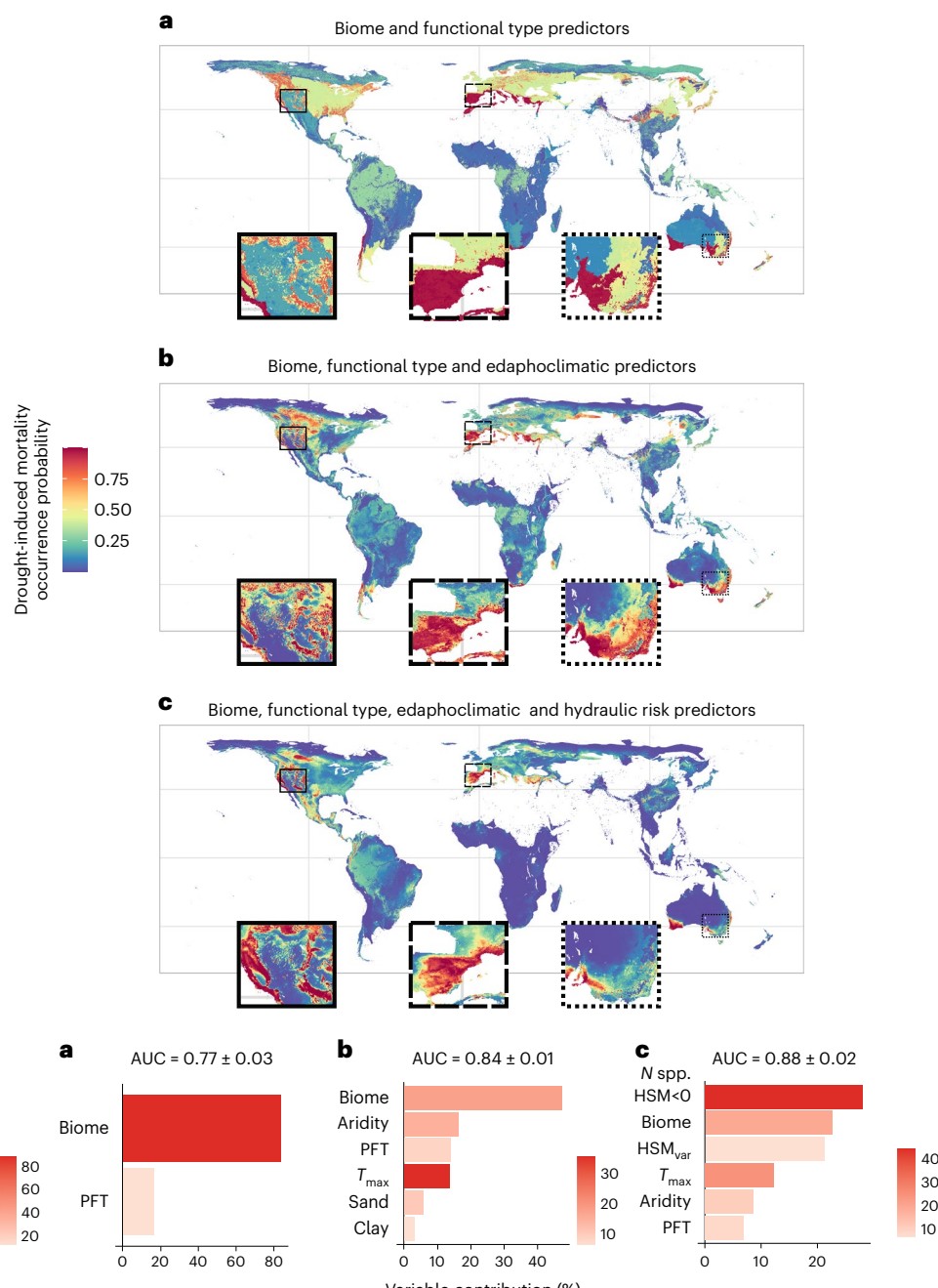

**Fig. 6 | Geographical projection of DIM occurrence probability, model's performance (test AUC), variables contribution and their permutation importance. a**–**c**, Predictors included in each case were: model type a, biome and PFT (**a**); model type a plus continuous edaphoclimatic variables (maximum temperature, $T_{max}$, aridity index, sand and clay content) (**b**); and model type b plus the number of species with HSM < 0 (*N* spp. HSM < 0) and HSM variance included in place of the two edaphoclimatic variables with the lowest contribution in model type b (sand and clay content), to keep the same number of predictive variables in model types b and model c (**c**).

Overall, sites comprising species assemblages with higher hydraulic risk (that is, lower mean and minimum HSM and higher number of species with HSM < 0) exhibited higher DIM probability. In the case of the relationship between DIM and the number of species with HSM < 0, the effect remained significant when species richness was included as a covariate. In fact, species richness itself was not a strong predictor of DIM. Thus, the relationship between the number of species with HSM < 0 and DIM was not driven by species number per se but by the relationship between DIM and the number of species with HSM < 0, expected to present a high hydraulic risk. We also show that places with higher HSM variability tend to present a higher DIM. This pattern was largely explained by the strong correlation between HSM variability and minimum HSM, the latter being strongly related to DIM probability. Our results show that the most hydraulically vulnerable species of an assemblage are strong indicators of site-specific mortality risks. Their removal could generate directional functional changes[51], decreasing site-specific HSM variability, negatively affecting functional diversity and potentially amplifying negative effects on ecosystem functioning[52,53].

## Predicting DIM occurrence
We built on our significant predictive models described above to estimate DIM occurrence probability worldwide using maximum entropy

models[35]. Our results supported the usefulness of the newly derived hydraulic risk metrics at the species-assemblage level to predict DIM, increasing predictive performance compared to models based only on edaphoclimatic variables, biome or PFTs (Fig. 6). The number of species with HSM < 0 was the most important explanatory variable in these models. Results showed that high DIM risk is predicted in, for example, the Mediterranean basin, southern Australia, western North America and western tropical South America. Models including hydraulic risk metrics better constrained DIM occurrence probability in places with abundant mortality information (for example, the Iberian Peninsula), limiting the environmental space where mortality is predicted to occur by considering the functional characterization of species assemblages. However, differences between models are more difficult to interpret in regions where mortality data are scarce or absent, such as the African continent and Russian boreal forests (Supplementary Fig. 3c). In these cases, the inclusion of hydraulic risk metrics may be overconstraining the model, leading to an underestimation of the probability of DIM.

These results show the potential of functional data to improve predictions of vegetation responses to climate change at broad spatial scales. By considering the geographical variability in functional composition, the physiological mechanisms involved in species responses to the environment are characterized and the vulnerability of plant communities can be better assessed.

### Limitations and future directions

Data on plant mortality occurrence and hydraulic traits are limited and may be subject to geographical, phylogenetic and ontogenetic biases[3,32,54]. However, in this study, we find similar patterns in the relationship between hydraulic risk and DIM across and within biomes. Thus, we posit that these relationships are not due just to a higher sampling of drier biomes but to a generalized pattern that is not expected to be an artefact of sampling bias. In any case, even with the most up-to-date hydraulics and mortality information, our results will need further confirmation in the future. Future efforts to improve the monitoring of observed DIM as well as the characterization of hydraulic risk under different climate change scenarios will enable better assessments of when and where high DIM is to be expected and the corresponding impacts on ecosystem composition, structure and function. Better knowledge on eco-evolutionary relationships among functional traits will improve predictive models, leading to lower imputation error and a better functional characterization of species assemblages.

The macro-evolutionary approach used in this study based on species presence–absence information also has limitations. The inclusion of intraspecific variability in future works will be very important to better assess geographical patterns in functional traits and associated environmental responses. Including data on species abundances will also lead to a more realistic characterization of the HSM distribution within each species assemblage. Results obtained here substantially differed from HSM projections using community weighted means for a smaller region (United States)[14], even though they were reasonably consistent for $P_{50}$ mean projections as well as for metrics that are not based on abundances, such as trait ranges (Supplementary Fig. 8). It is important to note that this study does not provide a causal explanation of DIM at the global scale. Instead, we show a relationship between functional composition, informed by phylogenetic position and edaphoclimatic variables, with DIM. Consequently, this relationship may also indicate an indirect relationship between hydraulic risk and mortality caused by environmental or phylogenetic signals.

In this study, HSM was considered a static proxy for hydraulic risk at a given site but any temporally explicit prediction of DIM risk would need to consider the characteristics of specific droughts in terms of duration and intensity and their impact on tissue-level exposure. Finally, considering additional ecological and historical factors such as changes in species-specific traits related to carbon metabolism, the likelihood of biotic attacks, extreme event legacies

and microclimatic conditions[22,38] should further improve predictions of DIM probability.

In conclusion, we show that species-assemblage hydraulic metrics are related to DIM and improve DIM prediction at the global scale. We show that locations with higher numbers of species with high hydraulic risk also have higher DIM. The approach presented here also represents a step forward in predicting plant functional trait values in vegetation, providing continuous maps that supplement environmental and coarse PFT characterizations. Further, the geographical characterization of functional trait distributions that we have provided here is probably of broad interest to improve the parameterization of terrestrial biosphere models[13,20] and complements other recent efforts using model inversion to predict hydraulic traits at the global scale[55]. Mortality estimates presented here are limited by the availability of spatially explicit hydraulic and mortality data as well as tree abundance data and should be seen as a starting point to improve global-scale mortality projections.

## Methods

### Species distribution data

Spatially explicit alpha-hull terrestrial range distributions of 44,901 species derived from compilations of species presence records[34] were used to determine species assemblages within 5 km grid cells. Species nomenclature was standardized using the Taxonstand R package[56] and species taxonomy was filled using the taxonlookup R package[57], both following The Plant List nomenclature.

### Hydraulic traits data

We extracted values from the recently updated xylem traits database[32] for minimum water potential recorded in the xylem ($P_{min}$) and water potential at the 50% and 88% loss of conductivity ($P_{50}$ and $P_{88}$) for 685, 1,376 and 735 species, respectively, measured in stems of mature individuals. The $P_{50}$ and $P_{88}$ included only observations with values <−0.5 MPa that originated from S-shaped vulnerability curves. Taxonomic standardization was carried out as described earlier.

The $P_{min}$ estimated as the absolute minimum xylem water pressure recorded for a given species can be prone to biases[54], so we tested for its relationship with soil minimum water availability and maximum vapour pressure deficit within the distribution of the species, which were considered to be among the main environmental drivers of its variation. The cross-species relationship between soil and plant minimum water potentials was positive and significant ($R^2 = 0.12$). The large scatter around this relationship probably reflects differences in rooting depth (and hence explored soil volume) across species, as well as substantial methodological uncertainties for both $P_{min}$ estimation approaches. The $P_{min}$ also showed a significant relationship with maximum vapour pressure deficit (VPD$_{max}$), as expected, with more negative minimum water potentials under a higher atmospheric water demand ($R^2 = 0.20$).

### Environmental data

To characterize edaphoclimatic affiliations for all the species for which we had range distributions, we downloaded global layers describing climatic variables from Worldclim[58] and soil characteristic variables from SoilGrids[59] at a resolution of 2.5 arcmin. We then extracted the values for each species using species range distributions data and the sf and raster R packages[60,61]. Edaphoclimatic variables were selected on the basis of their importance in a previous study[31]. The following layers describing species' historical climate (averaged values for 1970–2000)[58] were considered: mean annual temperature (°C), minimum temperature of the coldest month (°C), mean temperature of the wettest month (°C), mean temperature of the driest month (°C), isothermality (unitless), temperature seasonality (°C), annual precipitation (mm), precipitation of the wettest month (mm), precipitation of the driest month (mm), precipitation seasonality (mm), precipitation of the warmest quarter (mm), precipitation of the coldest quarter (mm),

mean solar radiation (kJ m$^{-2}$ d$^{-1}$), mean vapour pressure (kPa) and mean wind speed (m s$^{-1}$). We also extracted monthly maximum temperature values and the vapour pressure for the same months to calculate maximum vapour pressure deficit (kPa) for each species distribution using the SVP function from the humidity R package[62]. Layers describing soil characteristics were absolute depth to bedrock (cm), soil water content at 200 cm depth (percentage), cation exchange capacity at 30 cm depth (cmolc kg$^{-1}$, centimol positive charge per kg of soil), clay content at 30 cm depth (percentage), organic carbon at 30 cm depth (permille) and pH at 30 cm depth (pH).

Mean values for each species range were calculated for each edaphoclimatic variable and were transformed to achieve normality where needed (log- or square root-transformed). To summarize edaphoclimatic information, we implemented a principal component analysis on species mean values for the whole set of variables using the princomp function from the stats R package[63]. The first five principal components explained 82.3% of the variance and were used in further analyses.

Additional edaphoclimatic information required in some analyses (see the last two sections in Methods) was downloaded separately. This included the aridity index[64], historical maximum temperature for 1970–2000[58], as well as biome identity[65] and pixel-level PFT (ERA Copernicus 2019 land cover v.2.1.1)[66]. All these edaphoclimatic layers were aggregated to a 5 km$^2$ resolution for further use with the raster R package[61].

## Mortality database
We used a global database on forest die-off events related to drought and/or heat[1,3], which is an updated and geographically referenced version of the ref. [1] dataset. This new database was a spatial points data frame covering 1,303 mortality events records (Supplementary Fig. 3c), with documented affected species in each instance (>400 tree species worldwide). Taxonomic standardization was carried out for species in the mortality database as described above.

## Phylogenetic information
To include species phylogenetic information, we used a newly derived genus-level phylogeny covering 3,488 genera[67] to construct a phylogenetic distance matrix between taxa using the cophenetic.phylo function of the ape R package[68]. The distance matrix was used to calculate phylogenetic principal coordinates values for each genus using the pcoa function of the ape R package[68]. Then, coordinate values were assigned to each species[68]. Overall, we generated a dataset covering 44,901 species with complete edaphoclimatic and phylogenetic data and some sparse data on hydraulic traits distributed throughout the phylogeny. We also constructed a species-level phylogeny using the V.PhyloMaker R package[69] matching our species list. We used the species-level phylogeny only for plotting purposes because it contained many polytomies and because genus-level approaches can be considered more reliable, especially for tropical clades where species misidentification can be an issue[70].

## Hydraulic traits imputation
We used random-forest models as implemented by the missForest R package[33] to predict and impute species-level $P_{min}$ and $P_{50}$ values for the 44,901 woody plant species for which we had distribution data. This predictive framework was chosen on the basis of previous results that showed a strong relationship between these traits and edaphoclimatic and phylogenetic data[31]. Before performing the imputations, we tested the predictive performance of a set of models including different combinations of phylogenetic principal coordinates, edaphoclimatic principal components and including or excluding major evolutionary affiliation (angiosperms versus gymnosperms). We built models that predicted one trait at a time or both ($P_{min}$ and $P_{50}$), within the same model (in the latter case, trait covariation was explicitly considered).

To do so, we used the subset of species for which hydraulic measurements were available and calculated $R^2$ values following a tenfold cross-validation procedure using different proportions of train and test observations in each case (from 10% to 70% of data used to test and the remaining to train). Each model was iterated 100 times using a random selection of training and test points, maintaining the proportions in each case. We calculated the mean $R^2$ and its standard deviation in each case (Supplementary Table 1) and the model with the highest mean $R^2$ was subsequently used to predict trait values with all available data as training data and was iterated 100 times. The best predictive model included the first five phylogenetic principal coordinates and the first five edaphoclimatic principal components, while considering the covariation between traits and major evolutionary affiliation, reaching mean $R^2$ of 0.60 ± 0.10 and 0.54 ± 0.12 for $P_{min}$ and $P_{50}$, respectively (Supplementary Table 1; see Supplementary Fig. 9 for a schematic description of the methods). As some studies have pointed out that $P_{88}$ may be a better hydraulic failure threshold for angiosperm species[36], we also performed predictions using $P_{88}$ instead of $P_{50}$ for angiosperms ($P_{50/88}$ and HSM$_{50/88}$ hereafter).

Imputed values were summarized at the species level, calculating the mean and the standard deviation from the 100 iterations of the predictive model and HSM values were calculated from imputed mean-hydraulic trait values in each case (HSM = $P_{min} - P_{50}$). Imputed values were plotted on a species-level phylogeny (Fig. 1 shows hydraulic traits imputation at the species level for those species with at least one trait with observed values) as well as on the genus-level phylogeny (by averaging values per genera) (Supplementary Fig. 1 gives standard deviation of data aggregated at the genus level). To assess model uncertainty related to the identity and number of species used to train the predictive model, we repeated it 100 times, randomly excluding 20% of species with observed data each time and calculating the standard deviation of the predicted values for each species.

The predictive framework was also implemented using $P_{50}$ values for gymnosperm species and $P_{88}$ values for angiosperm species ($P_{50/88}$), calculating HSM$_{50/88}$ (Supplementary Fig. 2 gives a genus-level representation of these data). We obtained a lower predictive performance, reaching a mean $R^2$ of 0.43 ± 0.12 (mean and standard deviation for $P_{50/88}$ from the previously described cross-validation procedure), probably because of a higher error in $P_{88}$ estimates and lower data availability compared to $P_{50}$. Given the lower performance of HSM$_{50/88}$ models, the lower data availability for $P_{50/88}$ compared to $P_{50}$ and considering that $P_{88}$ was highly related to $P_{50}$ ($R^2 = 0.69$), we used $P_{50}$ and standard HSM to report the main results.

## Hydraulic metrics of species assemblages
To plot hydraulic metrics for species assemblages, we first spatially referenced species-level imputed data for 44,901 species using their spatial range distribution[34] (Supplementary Fig. 3a,b to see species range distribution coverage for imputed and observed traits data, respectively). Spatial projections were implemented by assuming fixed trait values at the species level (as we expect intraspecific variability to be much lower than interspecific variability for hydraulic traits)[71–73]. Then, we aggregated trait values for species with overlapping distributions at the pixel level by calculating their mean, minimum and variance as a measure of functional variability by using the fasterize function of the fasterize R package[74] and the rasterize function of the raster R package[61] in the case of the variance. By doing so, we obtained 5 km$^2$ raster layers for $P_{50}$ and HSM mean and their variability (Figs. 3a,b and 4a,b), minimum HSM (Fig. 3c), $P_{min}$ mean and its variability (Supplementary Fig. 4), $P_{50/88}$ and HSM$_{50/88}$ mean and their variability (Supplementary Fig. 5). Note that mean HSM and minimum HSM are reported as negative HSM so higher values represent higher hydraulic risk. This was performed for consistency with $P_{50}$ plots, as higher $P_{50}$ represents higher embolism vulnerability. For HSM and HSM$_{50/88}$ spatially referenced values, we also calculated the number of species with negative

values per pixel at 5 km$^2$ resolution using the same approach (Fig. 4c and Supplementary Fig. 5c for HSM and HSM$_{50/88}$, respectively). These maps should be interpreted as predicted values and then will only be relevant in areas with woody plant vegetation. However, we also provide maps excluding land cover categories without woody vegetation (using Copernicus, the land cover map previously referred to as a reference)[66] (Supplementary Fig. 10).

We also spatially aggregated cross-species $P_{50}$ and HSM standard deviations by calculating the mean from the 100 iterations of the predictive model including all species (Supplementary Fig. 7a,c,e) and excluding the 20% of species with observed trait data in each iteration (Supplementary Fig. 7b,d,f). Then, we report two measures of model uncertainty aggregated at the spatial scale: the first one showing the uncertainty of the predictive model at the species level and the second one the uncertainty linked to the identity of the species represented in the training data used. The uncertainty due to the identity of the species used to train models is higher than the model uncertainty (Supplementary Fig. 7).

To better visualize variability in raster plots, we restricted values using the clamp function from raster R package[61], setting the 0.05 quantile as the lower value and the 0.95 quantile as the upper value.

### Assessing the predictive capacity of hydraulic traits

First, we tested the relationship between imputed species-level HSM values and the presence–absence of observed mortality as well as the number of mortality events recorded per species as reported in the global mortality database[3]. We used generalized linear models through the glm function of the stats R package[63], setting the family parameter to 'binomial' in the first case and to 'Poisson' in the second one. To see the effects of angiosperm versus gymnosperm affiliation in this relationship, we included the major evolutionary affiliation as an explanatory factor interacting with HSM. As the number of species without observed mortality was much higher than the number with observed mortality, we randomly selected the same number of species without observed mortality events to match the number of species with mortality events (that is, 482). We repeated this procedure 100 times and averaged the results in both cases.

To explore the relationship between the spatial projection of hydraulic metrics and mortality occurrence as reported by the global DIM database[3], we used binomial generalized linear models with the glm function of the stats R package[63]. We kept one mortality event per square kilometre, reducing the number of geographical points with observed DIM from 1,303 to 882 to avoid over-representing areas with a higher sampling effort. To assess the degree of spatial autocorrelation of models, we performed Mantel tests on the residuals of all models using the function mantel.rtest of the Ade4r package[75]. The spatial autocorrelation was <0.06 in all cases. The response variable in our models was mortality occurrence (1 for pixels with at least one mortality event observed and 0 for the same number of randomly sampled pixels without observed mortality). Backgrounds could include some presences, so to deal with the lack of absence points we repeated models 100 times randomly changing background points and averaged results. The explanatory variables included HSM-derived variables related to the hydraulic risk of species assemblages (pixel mean, minimum, variance and number of species with HSM < 0), as well as their interaction with biome and PFT (for example, broadleaf deciduous, broadleaf evergreen, needle-leaved and so on) (Fig. 5 and Supplementary Fig. 6). An aridity index, annual precipitation and maximum temperature were also included as predictors in a separate model to assess their individual predictive power (Supplementary Table 2). Biome and functional type categories were reclassified to maintain as many observations per category as possible (Supplementary Table 4). We included biome and functional type as factors in the models to check for changes in the magnitude and direction of the relationships between species-assemblage hydraulic metrics and DIM as well as to improve predictions by better

representing broad vegetation types (for example, see the Amazon Basin in Fig. 6). Note that our data have a low number of observations in some biome and functional type groups, so no firm conclusions were drawn from the differences among factor levels.

The number of species per pixel was also included as a covariate in a model using the number of species with HSM < 0 to check for the effect of species number on its relationship with DIM occurrence. HSM variance and HSM minimum as well as their interaction were also considered together in the same model to better understand their non-independent relationship with DIM occurrence. Trend significance was tested by using the emmeans R package[76] (Supplementary Table 5). Each model was run 100 times using a different set of background points and pseudo-$R^2$ values were calculated using the rml R package[77] (Supplementary Table 2). Test AUC values were also calculated using the dismo R package[78] following a cross-validation procedure with 80% of the data to train and 20% to test. All models were rerun including aridity index values extracted from mortality and background points as a covariate to test whether trait effects remained significant when the climate was considered, which was the case. To check for variable significance, we implemented analysis of variance tests using the anova function from the stats R package[63] (Supplementary Table 3 gives the mean results calculated from 100 iterations in each case for models including aridity index as a covariate). As a further check, we repeated the same procedures but we were more restrictive in aggregating mortality data to avoid over-representing areas with higher sampling intensity (western United States, southwestern Australia and Europe)[79]. When we kept only one mortality occurrence per 10 km$^2$ (ref. 79), reducing the number of occurrences from 1,303 to 517, the results did not differ.

### Projecting mortality risk using maximum entropy models

We used maximum entropy models[35] as implemented by the dismo R package[78] to predict and project DIM risk at the global scale. We used this methodology instead of the previous binomial generalized linear models as it accounts better for presence/background point data under a predictive framework. This allowed us to better characterize the background by including more background points than presences, a procedure not recommended with generalized linear models[80]. Moreover, this technique presents higher predictive performance than generalized linear models because of its capability to account for nonlinearities and multiple interactions between predictors[81]. Three types of models were run: type a using only functional type and biome distributions as predictors, type b as in type a plus continuous edaphoclimatic variables and type c as in type b plus the projected hydraulic metrics as predictors. To maximize predictive performance while keeping the lowest number of predictors, only continuous variables with high predictive power that presented Pearson cross-correlation coefficients among themselves lower than 0.75 were included in models b and c. These variables were maximum temperature, aridity index, soil sand and clay content for models including edaphoclimatic variables and the number of species with HSM < 0, HSM variance, maximum temperature and aridity index for models including both hydraulic traits and edaphoclimatic variables. In all cases, biome and functional type were included as predictive factors. Note that none of the edaphoclimatic variables used to predict mortality was included in the edaphoclimatic principal components used to predict species-level hydraulic traits from which species-assemblage hydraulic metrics were calculated. Models b and c were constructed to contain the same number of predictors to facilitate their comparability.

In this instance, mortality data were aggregated to keep one occurrence per 10 km$^2$ to avoid overfitting[79] (number of occurrences 517) while standardizing the spatial resolution with the layers used as predictors. Models were trained using the 'hinge' option (similar to GAM) with 10,000 randomly sampled background points (but models were also trained using 1,000 and 50,000 randomly sampled background

points to assess model consistency). To evaluate model performance, each model was trained using 80% of the data and tested using the remaining 20% and this procedure was repeated 100 times in each case (randomly changing training and test data points) and test AUC values were calculated and summarized by calculating their mean and standard deviation to assess performance (Fig. 6). We made sure to include both points with observed mortality and background points in all cases by sampling the 80% and the 20% in each of these groups separately and then unifying the datasets, following previous implementations[82]. Finally, a single model trained using all observations was implemented for model types a, b and c (see earlier) and used to project mortality occurrence probability geographically (Fig. 6). Variable importance was assessed by its relative (percentage) contribution to the fit of the models as generated by the maxent jack-knife procedure, which compares the training gain for each variable in isolation to the training gain of the model with all variables (Fig. 6). Permutation importance was also calculated for each edaphoclimatic variable by randomly permuting presence and background values, re-evaluating the model and calculating the resulting drop in training AUC, normalized as a percentage (Fig. 6).

### Reporting summary

Further information on research design is available in the Nature Portfolio Reporting Summary linked to this article.

## Data availability

The minimum dataset needed to replicate the analyses can be found in the following public repository: https://doi.org/10.6084/m9.figshare.23635446.

## Code availability

The code used can be found in the following repository: https://github.com/pablosanchezmart/Sanchez-Martinez_etal-2022.

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

## Acknowledgements

This work was supported by the Spanish government via competitive grants FUN2FUN (no. CGL2013-46808-R), DRESS (no. CGL2017-89149-C2-1-R) and TRACES (no. PID2021-127452NB-I00) funded by MCIN/AEI/10.13039/501100011033; and by grant no. 2021 SGR 00849 funded by AGAUR. P.S.-M. acknowledges an FPU predoctoral fellowship from the Spanish Ministry of Science, Innovation and Universities (grant no. FPU18/04945). J.M.-V. benefited from an ICREA Academia award. R.A.S. is supported by Fondecyt-Iniciacion/2020 1100967, ANID, Chile and grant no. ANID/BASAL FB210006. J.-C.S. and W.-Y.G. thank the VILLUM FONDEN for support via VILLUM Investigator project 'Biodiversity Dynamics in a Changing World' (grant no. 16549). J.-C.S. also considers this work a contribution to Center for Ecological Dynamics in a Novel Biosphere (ECONOVO), funded by Danish National Research Foundation (grant no. DNRF173). J.-M.S.-D. acknowledges support from NASA grant no. 80NSSC 22K0883.

## Author contributions

P.S.-M., M.M., R.G.-V. and J.M.-V. designed the study. R.A.S. and K.G.D. provided the phylogeny. W.M.H. provided the hydraulics data. W.M.H. and C.A. provided the mortality data. J.-C.S., J.-M.S.-D. and W.-Y.G. provided the species range distribution data. P.S.-M. analysed data and generated the tables, figures and maps with input from M.M., J.M.-V. and R.G.-V. P.S.-M. wrote the first draft of the manuscript. All authors contributed substantially to revisions.

## Competing interests

The authors declare no competing interests.

## Additional information

**Correspondence and requests for materials** should be addressed to Pablo Sanchez-Martinez.

# Reporting Summary

## Statistics

For all statistical analyses, confirm that the following items are present in the figure legend, table legend, main text, or Methods section.

| n/a | Confirmed | |
|---|---|---|
| ☐ | ☒ | The exact sample size (*n*) for each experimental group/condition, given as a discrete number and unit of measurement |
| ☐ | ☒ | A statement on whether measurements were taken from distinct samples or whether the same sample was measured repeatedly |
| ☐ | ☒ | The statistical test(s) used AND whether they are one- or two-sided *Only common tests should be described solely by name; describe more complex techniques in the Methods section.* |
| ☐ | ☒ | A description of all covariates tested |
| ☐ | ☒ | A description of any assumptions or corrections, such as tests of normality and adjustment for multiple comparisons |
| ☐ | ☒ | A full description of the statistical parameters including central tendency (e.g. means) or other basic estimates (e.g. regression coefficient) AND variation (e.g. standard deviation) or associated estimates of uncertainty (e.g. confidence intervals) |
| ☐ | ☒ | For null hypothesis testing, the test statistic (e.g. *F*, *t*, *r*) with confidence intervals, effect sizes, degrees of freedom and *P* value noted *Give P values as exact values whenever suitable.* |
| ☐ | ☒ | For Bayesian analysis, information on the choice of priors and Markov chain Monte Carlo settings |
| ☐ | ☒ | For hierarchical and complex designs, identification of the appropriate level for tests and full reporting of outcomes |
| ☐ | ☒ | Estimates of effect sizes (e.g. Cohen's *d*, Pearson's *r*), indicating how they were calculated |

*Our web collection on statistics for biologists contains articles on many of the points above.*

## Software and code

Policy information about availability of computer code

| Data collection | Data used come from global databases which have been compiled by authors in previous works. More details on data measurement can be found in the methods section of the present work and the previously published ones describing the data used. The minimum dataset needed to replicate analyses can be found in DOI: 10.6084/m9.figshare.23635446. |
|---|---|
| Data analysis | The code for data analyses can be found in the following repository: https://github.com/pablosanchezmart/Sanchez-Martinez_etal-2022. |

For manuscripts utilizing custom algorithms or software that are central to the research but not yet described in published literature, software must be made available to editors and reviewers. We strongly encourage code deposition in a community repository (e.g. GitHub). See the Nature Portfolio guidelines for submitting code & software for further information.

## Data

Policy information about availability of data

All manuscripts must include a data availability statement. This statement should provide the following information, where applicable:
- Accession codes, unique identifiers, or web links for publicly available datasets
- A description of any restrictions on data availability
- For clinical datasets or third party data, please ensure that the statement adheres to our policy

The code used can be found in https://github.com/pablosanchezmart/Sanchez-Martinez_etal-2022. All data needed to replicate the analyses is available in previous

publications except for the hydraulics dataset, which will be published soon by William Hammond and collaborators in a data paper. The minimum dataset needed to replicate analyses can be found in DOI: 10.6084/m9.figshare.23635446.

## Human research participants

Policy information about studies involving human research participants and Sex and Gender in Research.

| | |
|---|---|
| Reporting on sex and gender | *Use the terms sex (biological attribute) and gender (shaped by social and cultural circumstances) carefully in order to avoid confusing both terms. Indicate if findings apply to only one sex or gender; describe whether sex and gender were considered in study design whether sex and/or gender was determined based on self-reporting or assigned and methods used. Provide in the source data disaggregated sex and gender data where this information has been collected, and consent has been obtained for sharing of individual-level data; provide overall numbers in this Reporting Summary.  Please state if this information has not been collected. Report sex- and gender-based analyses where performed, justify reasons for lack of sex- and gender-based analysis.* |
| Population characteristics | *Describe the covariate-relevant population characteristics of the human research participants (e.g. age, genotypic information, past and current diagnosis and treatment categories). If you filled out the behavioural & social sciences study design questions and have nothing to add here, write "See above."* |
| Recruitment | *Describe how participants were recruited. Outline any potential self-selection bias or other biases that may be present and how these are likely to impact results.* |
| Ethics oversight | *Identify the organization(s) that approved the study protocol.* |

Note that full information on the approval of the study protocol must also be provided in the manuscript.

# Field-specific reporting

Please select the one below that is the best fit for your research. If you are not sure, read the appropriate sections before making your selection.

☐ Life sciences       ☐ Behavioural & social sciences       ☒ Ecological, evolutionary & environmental sciences

For a reference copy of the document with all sections, see nature.com/documents/nr-reporting-summary-flat.pdf

# Ecological, evolutionary & environmental sciences study design

All studies must disclose on these points even when the disclosure is negative.

| | |
|---|---|
| Study description | Here, we use a newly global database to predict hydraulic risk of woody plant species using Random Forests. Then, we use generalized linear models to test whether hydraulic risk predict observed drought induced mortality (DIM). Finally, we project DIM occurrence probability worldwide using maximum entropy models. |
| Research sample | Hydraulics data comes from a global dataset on hydraulic traits which will be soon published by William Hammond. A previous version of this dataset can be found, for instance, in Sanchez-Martinez et al. 2020. Species distribution data comes from a compilation and generalization of global data on species occurrences (see Serra-Diaz et al. 2017). Mortality data comes from a global dataset compiling drought induced mortality occurrence (Hammond et al. 2022). All data sources are referenced in the main text. |
| Sampling strategy | In this global study, all available data that passed a quality filter (see methods) was used. |
| Data collection | Data collection information can be found in the cited papers which describe datasets used in this study. |
| Timing and spatial scale | Data used comes from peer reviewed publications reporting hydraulic traits, drought induced mortality and species distributions. Then, there is not a specific team scale. the spatial scale is global. |
| Data exclusions | For hydraulic traits, only data coming from stems of mature individuals were used. P50 and P88 included only observations with values < -0.5MPa that originated from S-shaped vulnerability curves, as these values are prone to present a high measurement error. |
| Reproducibility | No experiment was conducted in the present study, as it is a global analyses. |
| Randomization | No experiment was conducted in the present study, as it is a global analyses. |
| Blinding | No experiment was conducted in the present study, as it is a global analyses. |

Did the study involve field work?       ☐ Yes       ☒ No

# Reporting for specific materials, systems and methods

We require information from authors about some types of materials, experimental systems and methods used in many studies. Here, indicate whether each material, system or method listed is relevant to your study. If you are not sure if a list item applies to your research, read the appropriate section before selecting a response.

| Materials & experimental systems | | Methods | |
|---|---|---|---|
| **n/a** | **Involved in the study** | **n/a** | **Involved in the study** |
| ☒ ☐ | Antibodies | ☒ ☐ | ChIP-seq |
| ☒ ☐ | Eukaryotic cell lines | ☒ ☐ | Flow cytometry |
| ☒ ☐ | Palaeontology and archaeology | ☒ ☐ | MRI-based neuroimaging |
| ☒ ☐ | Animals and other organisms | | |
| ☒ ☐ | Clinical data | | |
| ☒ ☐ | Dual use research of concern | | |

