## [Peer Review File · Nature Ecology & Evolution]

Peer Review Information

Journal: Nature Ecology & Evolution

Manuscript Title: NATECOLEVOL-220716967C

Corresponding author name(s): Pablo Sanchez-Martinez

Editorial Notes:

Reviewer Comments & Decisions:

Decision Letter, initial version:

13th January 2023

Dear Dr Sanchez-Martinez,

Once again let me sincerely apologise for the length of time your manuscript has been under review. As mentioned in my previous emails, we were unable to obtain reports from two referees who originally agreed to review your manuscript -- one requested to withdraw due to personal circumstances, and another was unresponsive to our enquiries. In addition to this, we had to recruit an additional third referee to specifically comment on the machine learning methods, as both of the replacement referees stated that they were unable to assess this aspect of the manuscript. This third referee also requested several extensions to our originally agreed deadlines. I would like to assure you that instances such as this are very unusual in the review process, and we do understand that you will have hoped for a much faster review process than this.

However, I can now confirm that your Article entitled "Increased hydraulic risk in assemblages of woody plant species predicts spatial patterns of drought-induced mortality" has now been seen by 3 reviewers, whose comments are attached. In the light of their advice, we have unfortunately decided that we cannot offer to publish your manuscript in Nature Ecology & Evolution.

From the reports, you will see that while Referees #1 and #2 find your work of some potential interest, Referee #3 (an expert in traits, ecosystem ecology, distribution modelling, and machine learning) raises major concerns over the analysis approach used here, which they feel undermines the robustness of the conclusions and therefore overall strength of the novel conclusions that can be drawn at this stage.

While I appreciate that this will be extremely disappointing after this length of time, we feel that these specific criticisms of the validity of the methods and analysis from an expert in these approaches are sufficiently important that they cannot be overlooked, and as such must preclude publication of your work in Nature Ecology & Evolution.

I am sorry that we cannot be more positive on this occasion, but hope that you find the reviewers' comments helpful when preparing your paper for resubmission elsewhere.

[REDACTED]

Reviewer expertise:

Reviewer #1: Plant traits, hydraulics, drought

Reviewer #2: Plant traits, hydraulics

Reviewer #3: Traits, ecosystem ecology, distribution modelling, machine learning

Reviewers Comments:

Reviewer #1 (Remarks to the Author):

The manuscript entitled "Increased hydraulic risk in assemblages of woody plant species predicts spatial patterns of drought-induced mortality" had as main objective to analyze a global database on hydraulics traits and edaphoclimatic information, as well as detailed information about the region where the data were obtained, in order to achieve a robust model to predict drought-induced mortality (DIM) of woody plants. The topic addressed in the manuscript is very interesting and current, especially with climate change causing great effects on assemblages of woody species on all continents. The authors devoted greater attention to the hydraulic risk presented by the species and thus arrive at a biogeographical pattern.

I agree that the sample size and the tools used by the authors are adequate and also allow reaching the conclusion presented in the text.

On the other hand, it is undeniable that with increasingly severe and prolonged drought events, changes in carbon metabolism, especially NSC, have an impact on the survival of woody species. In addition, the authors are knowledgeable enough to know that the relationship between carbon metabolism and water status in woody species is inseparable. Roots do not photosynthesize and depend on the supply of NSC to keep working (uptake water and mineral nutrients). I suggest that the authors consider this aspect in a little more detail in this text. References like these are a good start for them to introduce a topic:

McDowell, N., Pockman, W.T., Allen, C.D., Breshears, D.D., Cobb, N., Kolb, T. et al. (2008) Mechanisms of plant survival and mortality during drought: why do some plants survive while others succumb to drought? *The New Phytologist*, 178, 719–739

Munné-Bosch, S. & Alegre, L. (2004) Die and let live: leaf senescence contributes to plant survival under drought stress. *Functional Plant Biology*, 31, 203–216

Sala, A., Piper, F. & Hoch, G. (2010) Physiological mechanisms of drought induced tree mortality are far from being resolved. *The New Phytologist*, 186, 274–281.

Sala, A., Woodruff, D.R. & Meinzer, F.C. (2012) Carbon dynamics in trees: feast or famine? *Tree Physiology*, 32, 764–775

Santos, M. Nicodemos, J. and Santos, M.G. (2022) Dynamics of nonstructural carbohydrates in a deciduous woody species from tropical dry forests under recurrent water deficit. *Physiologia Plantarum*. Doi: 10.1111/ppl.13632

2Reviewer #2 (Remarks to the Author):

This manuscript is well written, interesting, and timely. Climate change is causing woody plant assemblages to experience drought-induced mortality worldwide leading to changes in ecosystem composition, function, and structure. Previous studies have shown that hydraulic traits can improve our understanding and our capacity to predict drought-induced mortality. Plant hydraulic traits and strategies are starting to be incorporated into different vegetation models and assessments of forest vulnerability but the availability of hard to measure hydraulic traits (e.g., P50) that have been shown to have the best predictive power of drought-induced mortality have been limited until now. This is where this study comes into play by incorporating a large recently updated hydraulic trait database extracted from previously published studies.

The authors incorporate species phylogenetic data, species estimated edaphoclimatic affiliations, and the hydraulic trait dataset into random forest models to predict Pmin, P50, and hydraulic safety margins. They then georeferenced the predicted hydraulic traits by using species distribution models to create aggregated maps of hydraulic traits of species assemblages at the local and global scale. The species assemblage mean HSM, minimum HSM, HSM variability, and the number of species with negative HSM were then used in linear models to predict drought-induced mortality (DIM) from a previously georeferenced dataset of observations of DIM. The species assemblage mean HSM, minimum HSM, HSM variability, and the number of species with <0 HSM were also used in maxent models with different environmental predictors to project the probability of DIM worldwide.

The main findings of this study are:

- Overall, most species have small HSM (< 1 MPa), which aligns with previous findings and supports the hypothesis that species prioritize carbon fixation at the expense of increased hydraulic risk.
- The authors did not find a significant association between records of DIM occurrence and species-level HSMs, but they did find that species with larger HSM suffered from a lower number of DIM events although the R^2 was only 0.14. The authors conclude that even though species with low HSM tend to present a higher number of DIM events, this information is not sufficient to predict the probability that a species will suffer DIM.
- Using the aggregated hydraulic trait imputations for species assemblages in 5 by 5 km grid cells, they found that areas with high drought incidence had lower mean P50 but not higher HSMs. The authors conclude that species converged towards similar HSMs independent of drought exposure.
- There was higher species assemblage variability in P50 and HSM in locations with higher drought incidence, a finding that confirms previous findings from regional-level studies that showed that hydraulic trait variability is higher in locations with water limitations.
- The authors found that the model projections showed species assemblages with a high number of species with HSM below zero occurred both in dry and wet places, but that the lowest HSMs were found in places with high drought incidence.
- They found that species-assemblage HSMs were better predictors of DIM than species-level HSMs ($R^2 = 0.36$ versus $R^2 = 0.14$). Sites with species assemblages with lower mean and minimum HSM and a higher number of species with HSM < 0 had a higher probability of suffering from DIM. As to be expected, the authors find that most hydraulically vulnerable species of an assemblage are expected to be under a higher mortality risk.

3-The authors found that incorporating hydraulic risk metrics at the species assemblage level increased the performance of the models at estimating the probability of the occurrence of DIM, in comparison with the models that were based only on environmental variables. They found that the number of species with HSM < 0 was the most important explanatory variable in the models. Although, the authors highlight that the predictive power of the models is better for locations for which there are more records of DIM.

The value of this study lies in that it integrates a recently updated global map of georeferenced events of drought-induced mortality (Hammond et al. 2022, Nat. Commun.) with a large global dataset of plant hydraulic traits extracted from existing literature to increase the predictive power of models at estimating the probability of DIM.

The most important features of this manuscript are that the authors found that incorporating hydraulic risk metrics at the species assemblage level increased the performance of the models at estimating the probability of the occurrence of DIM and that the number of species with HSM < 0 was the most important explanatory variable in the models. I believe the results of this study to be of interest to people in the field of plant hydraulics, ecophysiology, and people interested in drivers of forest mortality and how to predict DIM.

I do not have any concerns regarding flaws in the data used, statistical analyses, or interpretation of the results and conclusion, all are appropriate and well presented. My only suggestion for improvement of the manuscript is that in figure 2a I suggest flipping the scale so that areas with more negative P50 values are represented in green blue to match how the color scheme has been used in Figure 2b. It seems more intuitive that more negative P50 values should be represented with the same color as larger HSMs.

Reviewer #3 (Remarks to the Author):

The paper titled "Increased hydraulic risk in assemblages of woody plant species predicts spatial patterns of drought-induced mortality" by Pablo et al. mainly integrates information on species' edaphoclimatic niches, phylogeny, and hydraulic traits to estimate the hydraulic risk of woody plants globally using machine learning. The topic is interesting, but the conclusions are not novel, and not robust as well.

Here are my main comments:

First, the authors used the random forest to establish the relationship between phylogenetic data jointly with edaphoclimatic affiliations and trait covariation predicted species-specific Pmin and P50, and upscale the P50 and HSW from the individual level to the global scale. I doubt that because there is no species composition map on grid-scale and authors only used some plots in boreal regions of Asia. Furthermore, the authors are very ambitious to provide the spatial HSM map, even for the human-disturbed regions (e.g., croplands and urban)? Why does this make sense if you do not have samplings?

4Second, regarding the global HSM map, the patterns largely follow some climatic variables (for example, Fig. 2f), and the map did not show reasonable heterogeneity. I do not know how to use this if the minimum HSW in the whole Amazon forest is just one value.

Third, when the authors talk about drought-induced mortality, does that mean grassland, shrubland, or forests? I guess most of them should be forests (maybe I am wrong). However, this paper just simply compared the mortality and HSM in their paper without considering the difference between forests and other ecosystems. Furthermore, drought duration and drought intensity both affect vegetation mortality. For example, for the species with high hydraulic tolerance, server drought events are also able to kill the trees, however, the paper did not talk about that.

Overall, the authors should deeply think about their work and focus on some small regions to study the relations between hydraulic risk and forest mortality first rather than on a global scale.

** *Communications Biology* is a selective Nature Portfolio title publishing Open Access research that brings new insight in all areas of the biological sciences. [Additional journal metrics and information can be found here](https://www.nature.com/commsbio/journal-information/journal-impact). Their editors prioritise good author service, fast peer review (in 2021, the median time to decision after first review was 40 days), and are happy to answer any questions you may have [\(commsbio@nature.com\)](mailto:commsbio@nature.com). The journal has an Impact Factor of 6.548, a CiteScore of 6.0 and a Scimago quartile ranking of Q1.

Please note that *Communications Biology* is a fully open-access journal and an article processing charge will apply to any papers accepted for publication. Our [open access pages](https://www.nature.com/commsbio/about/open-access) contain information about article processing charges, open access funding, and advice and support from Springer Nature.

If you wish to transfer your manuscript to *Communications Biology*, please use our manuscript transfer portal using the link below to initiate the transfer to this journal (or to another journal of your choice in the Nature Research portfolio). If you transfer to Nature-branded journals or to the Communications journals, you will not have to re-supply manuscript metadata and files. This link can only be used once and remains active until used. For more information, please see our [manuscript transfer FAQ](https://www.nature.com/nature-portfolio/for-authors/transfer) page.

** *Communications Biology* will send your work for peer review if you choose to transfer. *Communications Biology* is a selective Nature Portfolio title publishing Open Access research

that brings new insight in all areas of the biological sciences. [Additional journal metrics and information can be found here](https://www.nature.com/commsbio/journal-information/journal-impact). Their editors prioritise good author service, fast peer review (in 2021, the median time to decision after first review was 40 days), and are happy to answer any questions you may have [at commsbio@nature.com](mailto:commsbio@nature.com). The journal has an Impact Factor of 6.548, a CiteScore of 6.0 and a Scimago quartile ranking of Q1.

Please note that *Communications Biology* is a fully open-access journal and an article processing charge will apply to any papers accepted for publication. Our [open access pages](https://www.nature.com/commsbio/about/open-access) contain information about article processing charges, open access funding, and advice and support from Springer Nature.

You may transfer your manuscript to *Communications Biology*, please use our manuscript transfer portal using the link below to initiate the transfer to this journal (or to another journal of your choice in the Nature Research portfolio). If you transfer to Nature-branded journals or to the Communications journals, you will not have to re-supply manuscript metadata and files. This link can only be used once and remains active until used. For more information, please see our [manuscript transfer FAQ](https://www.nature.com/nature-portfolio/for-authors/transfer) page.

**

** For Nature Research general information and news for authors, see <http://npg.nature.com/authors>.

Decision Letter, first revision:

8th February 2023

Dear Dr Sanchez-Martinez,

Thank you for your letter asking us to reconsider our decision on your Article entitled "Increased hydraulic risk in assemblages of woody plant species predicts spatial patterns of drought-induced mortality". After careful consideration we have decided that we would be willing to consider a revised version of your manuscript.

Along with your revised manuscript, you should also submit a separate point-by-point response to all of the concerns raised by the reviewers, in each case describing what changes have been made to the manuscript or, alternatively, if no action has been taken, providing a compelling argument for why that is the case. If we feel that a substantial attempt has been made to address the reviewers' comments, this response will be sent back to the reviewers - along with the revised manuscript - so

6that they can judge whether their concerns have been addressed satisfactorily or otherwise.

I should stress, however, that we would be reluctant to trouble our reviewers again unless we thought that their comments had been addressed in full.

- ensure it complies with our format requirements for Articles as set out in our guide to authors at www.nature.com/natecolevol/authors/index.html

- state in a cover note the length of the text, methods and legends; the number of references and the number of display items.

Please ensure that all correspondence is marked with your Nature Ecology & Evolution reference number in the subject line.

Please use the following link to submit your revised manuscript:

[REDACTED]

I would appreciate it if you could tell me if you think you will be able to submit a revised manuscript, and also the likely timescale.

I look forward to hearing from you soon.

[REDACTED]

Author Rebuttal, first revision:

Comments from the reviewers:

Reviewer #1 (Remarks to the Author):

The manuscript entitled “Increased hydraulic risk in assemblages of woody plant species predicts spatial patterns of drought-induced mortality” had as main objective to analyze a global database on hydraulics traits and edaphoclimatic information, as well as detailed information about the

7region where the data were obtained, in order to achieve a robust model to predict drought-induced mortality (DIM) of woody plants. The topic addressed in the manuscript is very interesting and current, especially with climate change causing great effects on assemblages of woody species on all continents. The authors devoted greater attention to the hydraulic risk presented by the species and thus arrive at a biogeographical pattern.

I agree that the sample size and the tools used by the authors are adequate and also allow reaching the conclusion presented in the text. On the other hand, it is undeniable that with increasingly severe and prolonged drought events, changes in carbon metabolism, especially NSC, have an impact on the survival of woody species. In addition, the authors are knowledgeable enough to know that the relationship between carbon metabolism and water status in woody species is inseparable. Roots do not photosynthesize and depend on the supply of NSC to keep working (uptake water and mineral nutrients). I suggest that the authors consider this aspect in a little more detail in this text. References like these are a good start for them to introduce a topic:

McDowell, N., Pockman, W.T., Allen, C.D., Breshears, D.D., Cobb, N., Kolb, T. et al. (2008) Mechanisms of plant survival and mortality during drought: why do some plants survive while others succumb to drought? *The New Phytologist*, 178, 719–739

Munné-Bosch, S. & Alegre, L. (2004) Die and let live: leaf senescence contributes to plant survival under drought stress. *Functional Plant Biology*, 31, 203–216

Sala, A., Piper, F. & Hoch, G. (2010) Physiological mechanisms of drought induced tree mortality are far from being resolved. *The New Phytologist*, 186, 274–281.

Sala, A., Woodruff, D.R. & Meinzer, F.C. (2012) Carbon dynamics in trees: feast or famine? *Tree Physiology*, 32, 764–775

Santos, M. Nicodemos, J. and Santos, M.G. (2022) Dynamics of nonstructural carbohydrates in a deciduous woody species from tropical dry forests under recurrent water deficit. *Physiologia Plantarum*. Doi: 10.1111/ppl.13632

We thank the reviewer for their positive assessment and suggestions. We agree with the reviewer on the need to include more detail regarding carbon starvation as an alternative process to hydraulic dysfunction, and the linkage between these processes leading to mortality. We have included some discussion in this regard in L151-164 citing McDowell et al. (2008) and Sala, Woodruff & Meinzer (2012) and McDowell et al. (2022), as the reviewer suggested. We further mentioned the carbon metabolism in the future directions section (L272-273),

Reviewer #2 (Remarks to the Author):

This manuscript is well written, interesting, and timely. Climate change is causing woody plant assemblages to experience drought-induced mortality worldwide leading to changes in ecosystem composition, function, and structure. Previous studies have shown that hydraulic traits can improve our understanding and our capacity to predict drought-induced mortality. Plant hydraulic traits and strategies are starting to be incorporated into different vegetation models and assessments of forest vulnerability but the availability of hard to measure hydraulic traits (e.g., P50) that have been shown to have the best predictive power of drought-induced mortality have been limited until now. This is where this study comes into play by incorporating a large recently updated hydraulic trait database extracted from previously published studies.

The authors incorporate species phylogenetic data, species estimated edaphoclimatic affiliations, and the hydraulic trait dataset into random forest models to predict Pmin, P50, and hydraulic safety margins. They then georeferenced the predicted hydraulic traits by using species distribution models to create aggregated maps of hydraulic traits of species assemblages at the

10local and global scale. The species assemblage mean HSM, minimum HSM, HSM variability, and the number of species with negative HSM were then used in linear models to predict drought-induced mortality (DIM) from a previously georeferenced dataset of observations of DIM. The species assemblage mean HSM, minimum HSM, HSM variability, and the number of species with <0 HSM were also used in maxent models with different environmental predictors to project the probability of DIM worldwide.

The main findings of this study are:

-Overall, most species have small HSM (< 1 MPa), which aligns with previous findings and supports the hypothesis that species prioritize carbon fixation at the expense of increased hydraulic risk.

-The authors did not find a significant association between records of DIM occurrence and species-level HSMs, but they did find that species with larger HSM suffered from a lower number of DIM events although the R^2 was only 0.14. The authors conclude that even though species with low HSM tend to present a higher number of DIM events, this information is not sufficient to predict the probability that a species will suffer DIM.

-Using the aggregated hydraulic trait imputations for species assemblages in 5 by 5 km grid cells, they found that areas with high drought incidence had lower mean P50 but not higher HSMs.

The authors conclude that species converged towards similar HSMs independent of drought exposure.

-There was higher species assemblage variability in P50 and HSM in locations with higher drought incidence, a finding that confirms previous findings from regional-level studies that showed that hydraulic trait variability is higher in locations with water limitations.

-The authors found that the model projections showed species assemblages with a high number of species with HSM below zero occurred both in dry and wet places, but that the lowest HSMs were found in places with high drought incidence.

-They found that species-assemblage HSMs were better predictors of DIM than species-level HSMs ($R^2 = 0.36$ versus $R^2 = 0.14$). Sites with species assemblages with lower mean and minimum HSM and a higher number of species with $HSM < 0$ had a higher probability of suffering from DIM. As to be expected, the authors find that most hydraulically vulnerable species of an assemblage are expected to be under a higher mortality risk.

-The authors found that incorporating hydraulic risk metrics at the species assemblage level increased the performance of the models at estimating the probability of the occurrence of DIM, in comparison with the models that were based only on environmental variables. They found that the number of species with $HSM < 0$ was the most important explanatory variable in the models.

Although, the authors highlight that the predictive power of the models is better for locations for which there are more records of DIM.

The value of this study lies in that it integrates a recently updated global map of georeferenced events of drought-induced mortality (Hammond et al. 2022, Nat. Commun.) with a large global dataset of plant hydraulic traits extracted from existing literature to increase the predictive power of models at estimating the probability of DIM.

The most important features of this manuscript are that the authors found that incorporating hydraulic risk metrics at the species assemblage level increased the performance of the models at estimating the probability of the occurrence of DIM and that the number of species with HSM < 0 was the most important explanatory variable in the models. I believe the results of this study to be of interest to people in the field of plant hydraulics, ecophysiology, and people interested in drivers of forest mortality and how to predict DIM. I do not have any concerns regarding flaws in the data used, statistical analyses, or interpretation of the results and conclusion, all are appropriate and well presented. My only suggestion for improvement of the manuscript is that in figure 2a I suggest flipping the scale so that areas with more negative P50 values are represented in green blue to match how the color scheme has been used in Figure 2b. It seems more intuitive that more negative P50 values should be represented with the same color as larger HSMs.

We want to thank the reviewer for his/her detailed review and a very clear elucidation of the main novel results of our work. The scale of figure 2 has been changed following the reviewer suggestion, so more negative P_{50} mean values (Figure 2b) correspond now to lower hydraulic risk (Figure 2c). Consistently, the scale of figure 2g has also been flipped so higher values represent higher maximum hydraulic risk. As this paper is focused on hydraulic risk, we thought that it would make more sense to represent high values to represent high vulnerability (P_{50}) to embolism and high hydraulic risk (-HSM) in figure 2 maps. Then, we flipped HSM scale instead of P_{50} scale, which we think leads to an easier and more intuitive comparison between P_{50} and hydraulic risk. This is explained in the current version of the text (L415-417) and in figure 2 caption. We thank the reviewer for its contribution, as we think that the interpretation of figure 2 is now clearer.

Reviewer #3 (Remarks to the Author):

The paper titled “Increased hydraulic risk in assemblages of woody plant species predicts spatial patterns of drought-induced mortality” by Pablo et al. mainly integrates information on species’ edaphoclimatic niches, phylogeny, and hydraulic traits to estimate the hydraulic risk of woody plants globally using machine learning. The topic is interesting, but the conclusions are not novel, and not robust as well.

We thank the reviewer for their careful review and assessment. First, we will address the reviewer’s concern regarding the novelty of our study’s conclusions. To highlight the novelty of our study, we have clarified the implications of the main results in the text, so they are more evident to a general audience (L247-250, L277-278). To summarise here, this is the first study that integrates species distributions with hydraulic traits of woody plants to calculate hydraulic risk metrics for species assemblages. Then, we show for the first time how species assemblage hydraulic risk is positively related to the occurrence of drought induced mortality (DIM) at a

15global scale, and that including these hydraulic attributes improves the capacity to predict the occurrence of DIM relative to models using only environmental predictors and biome/functional type information. Its novelty resides both in the newly developed methodology and the globally distributed data synthesized here for the first time, as well as in the conclusions drawn from them. Second, regarding the reviewer's claim that our study is not robust, we acknowledge that our prior manuscript did not clearly define important steps in our methods, and their addition (as detailed in the specific comments below) demonstrates the robust nature of our approach.

Here are my main comments:

First, the authors used the random forest to establish the relationship between phylogenetic data jointly with edaphoclimatic affiliations and trait covariation predicted species-specific Pmin and P50, and upscale the P50 and HSW from the individual level to the global scale. I doubt that because there is no species composition map on grid-scale and authors only used some plots in boreal regions of Asia.

We appreciate the reviewer drawing our attention to a lack of clarity in our description of the methodology. We have strived to improve clarity in this revised version. The relationship between traits, their covariation and phylogenetic and edaphoclimatic data was established in a previous study (Sanchez-Martinez et al., 2020). This reference was already cited in the previous

version, but we have included it also in the part describing the random forest methods so that readers can refer to this previous work to better understand the relationship between these traits and phylogenetic and environmental data (L365-366). In the present work, we used these previously described relationships to predict species-level traits for unsampled species by means of random forest models (see “Hydraulic traits predictions using phylogenetic and edaphoclimatic data” in the methods). Then, we aggregated imputed data for species assemblages using the most comprehensive dataset on woody species range distributions ever compiled, which is plotted in fig. S4 and referenced (Serra-Diaz et al., 2017) (see “Species distribution data” and “Geographical projection of hydraulic traits and calculation of hydraulic metrics of species assemblages” of methods). This dataset consists of range distributions for more than 45,000 species based on a generalization of species presence records. We do not use data from plots as suggested by the reviewer, and the fact that boreal areas present a lower number of data points is due to their lower woody plant species richness (Figure S4). Figure S11, showing a schematic description of the methodology implemented, has been improved to further clarify the methods used. We acknowledge that data availability may also be a limiting factor in driving some of our results especially in boreal forests of Russia. In the current version of the manuscript, we specifically refer to the data availability issue (L180-182, 195-197) and we have further referred to the study and data limitations in the “limitations and future directions” section (L252-254, 261-262, 269-271).

Furthermore, the authors are very ambitious to provide the spatial HSM map, even for the human-disturbed regions (e.g., croplands and urban)? Why does this make sense if you do not have samplings?

We think the reviewer may have misinterpreted our methods, and we apologise for the lack of clarity in our previous version of the manuscript. In this revised version, we try and provide an improved description of our methods. We based our study on modelled range distributions of woody plant species (see previous comment). Therefore, the grid cell predictions are not meant to apply to all land cover within the grid cell but should only be interpreted for its tree-dominated land covers, as done, for instance, when modelling climatic suitability and species distributions. We have aimed to include clarification on this issue in the revised version (L417-419). Because of the scale of the analysis, most pixels include woody vegetation. Moreover, none of the DIM events in the dataset used occur in croplands or urban areas (Hammond et al., 2022). However, to address this comment we have applied a mask on our results to show only projections for those land covers which present woody plants (according to Copernicus land cover layer). These maps are shown now in Figure S12 and referred to in lines 419-420).

Second, regarding the global HSM map, the patterns largely follow some climatic variables (for example, Fig. 2f), and the map did not show reasonable heterogeneity. I do not know how to use this if the minimum HSW in the whole Amazon forest is just one value.

We thank the reviewer for raising this important issue. We believe that the lack of variability in minimum values is itself an interesting result, as it may indicate that minimum values are mainly determined by widespread species within a region that present low HSM values. In the current version of the manuscript, we further discuss this aspect in response to this comment, while mentioning the potential effect of data availability on this result (L192-197). We also included additional discussion regarding the interpretation of the maps (L177-182, L198-202) and included a visual example in figure 2 to better elucidate how we obtained global maps from the functional composition of species assemblages, and thus facilitate the interpretation of the figure.

Third, when the authors talk about drought-induced mortality, does that mean grassland, shrubland, or forests? I guess most of them should be forests (maybe I am wrong). However, this paper just simply compared the mortality and HSM in their paper without considering the difference between forests and other ecosystems. Furthermore, drought duration and drought intensity both affect vegetation mortality. For example, for the species with high hydraulic tolerance, severe drought events are also able to kill the trees, however, the paper did not talk about that.

The reviewer raises a topic that we failed to sufficiently clarify in the previous version of the manuscript. We better understand now that the text may have not been clear enough in that we are referring to woody plant species only. We have addressed this comment by explicitly stating

the focus on woody plant species in several sections of the manuscript (e.g., L35, L105 and L418), as well as the title and the abstract. We also considered differences between biome and functional type distributions when testing for the relationship between hydraulic risk metrics and mortality occurrence (Figure 3, Figure S7). We agree with the reviewer that including some more discussion on drought duration vs intensity may be useful, and we have thus included it in the current version (L161-164, L269-271). However, we would like to note that both drought duration and intensity are included (at least conceptually) in minimum water potential (Bhaskar & Ackerly, 2006; Martínez-Vilalta et al., 2021). In addition, we show how highly resistant species may also experience hydraulic dysfunction (high hydraulic risk), addressing the reviewer's comment on this issue (L142-145).

Overall, the authors should deeply think about their work and focus on some small regions to study the relations between hydraulic risk and forest mortality first rather than on a global scale. We appreciate this perspective. However, in this study, we are focusing on a global scale and take advantage of a recent increase in data availability on hydraulic traits, DIM occurrence, and woody species distributions, which allow us to generalize regional functional patterns and relate them to species composition and mortality occurrence. Regional and local studies are referenced in the manuscript (Anderegg et al., 2015; García-Valdés et al., 2021; Rowland et al., 2015). Importantly, part of the novelty and interest of this study is the fact that it is performed at the global scale. To clarify this issue, additional discussion on the implications of this study has now

been included (L247-250). Moreover, we have specifically added some discussion on the limitations in the last section, which now reads: “limitations and future directions”. We hope that these changes will clarify the novelty and implications of the current study for a general audience.

Literature cited

- Anderegg, W. R. L., Flint, A., Huang, C. Y., Flint, L., Berry, J. A., Davis, F. W., Sperry, J. S., & Field, C. B. (2015). Tree mortality predicted from drought-induced vascular damage. *Nature Geoscience*, 8(5), 367–371. <https://doi.org/10.1038/ngeo2400>
- Bhaskar, R., & Ackerly, D. D. (2006). Ecological relevance of minimum seasonal water potentials. *Physiologia Plantarum*, 127(3), 353–359. <https://doi.org/10.1111/j.1399-3054.2006.00718.x>
- Butler, E. E., Datta, A., Flores-Moreno, H., Chen, M., Wythers, K. R., Fazayeli, F., Banerjee, A., Atkin, O. K., Kattge, J., Amiaud, B., Blonder, B., Boenisch, G., Bond-Lamberty, B., Brown, K. A., Byun, C., Campetella, G., Cerabolini, B. E. L., Cornelissen, J. H. C., Craine,

- J. M., ... Schlesinger, W. H. (2017). Mapping local and global variability in plant trait distributions. *Proceedings of the National Academy of Sciences of the United States of America*, 114(51), E10937–E10946. <https://doi.org/10.1073/pnas.1708984114>
- García-Valdés, R., Vayreda, J., Retana, J., & Martínez-Vilalta, J. (2021). Low forest productivity associated with increasing drought-tolerant species is compensated by an increase in drought-tolerance richness. *Global Change Biology*, June 2020, 1–15. <https://doi.org/10.1111/gcb.15529>
- Hammond, W. M., Williams, A. P., Abatzoglou, J. T., Adams, H. D., & Klein, T. (2022). Global field observations of tree die-off reveal hotter-drought fingerprint for Earth ' s forests. *Nature Communications*.
- Martínez-Vilalta, J., Santiago, L. S., Poyatos, R., Badiella, L., de Cáceres, M., Aranda, I., Delzon, S., Vilagrosa, A., & Mencuccini, M. (2021). Towards a statistically robust determination of minimum water potential and hydraulic risk in plants. In *New Phytologist*. <https://doi.org/10.1111/nph.17571>
- Rowland, L., da Costa, A. C. L., Galbraith, D. R., Oliveira, R. S., Binks, O. J., Oliveira, A. A. R., Pullen, A. M., Doughty, C. E., Metcalfe, D. B., Vasconcelos, S. S., Ferreira, L. V., Malhi, Y., Grace, J., Mencuccini, M., & Meir, P. (2015). Death from drought in tropical forests is triggered by hydraulics not carbon starvation. *Nature*, 1–13. <https://doi.org/10.1038/nature15539>

Sanchez-Martinez, P., Martínez-Vilalta, J., Dexter, K. G., Segovia, R. A., & Mencuccini, M.

(2020). Adaptation and coordinated evolution of plant hydraulic traits. *Ecology Letters*, 23(11), 1599–1610. <https://doi.org/10.1111/ele.13584>

Serra-Diaz, J. M., Enquist, B. J., Maitner, B., Merow, C., & Svenning, J. C. (2017). Big data of tree species distributions: how big and how good? *Forest Ecosystems*, 4(1), 0–12.

<https://doi.org/10.1186/s40663-017-0120-0>

Swenson, N. G., Enquist, B. J., Pither, J., Kerkhoff, A. J., Boyle, B., Weiser, M. D., Elser, J. J.,

Fagan, W. F., Forero-Montaña, J., Fyllas, N., Kraft, N. J. B., Lake, J. K., Moles, A. T.,

Patiño, S., Phillips, O. L., Price, C. A., Reich, P. B., Quesada, C. A., Stegen, J. C., ...

Nolting, K. M. (2012). The biogeography and filtering of woody plant functional diversity in North and South America. *Global Ecology and Biogeography*, 21(8), 798–808.

<https://doi.org/10.1111/j.1466-8238.2011.00727.x>

Trugman, A. T., Anderegg, L. D. L., Shaw, J. D., & Anderegg, W. R. L. (2020). Trait velocities

reveal that mortality has driven widespread coordinated shifts in forest hydraulic trait composition. *Proceedings of the National Academy of Sciences*, 201917521.

<https://doi.org/10.1073/pnas.1917521117>

Decision Letter, second revision:

13th June 2023

Dear Dr. Sanchez-Martinez,

Thank you for submitting your revised manuscript "Increased hydraulic risk in assemblages of woody plant species predicts spatial patterns of drought-induced mortality" (NATECOLEVOL-220716967B), and thank you again for your patience while we sought a replacement reviewer for Referee #3. I can confirm that it has now been seen again by one of the original reviewers, together with a new reviewer (Referee #4) who has been asked to assess the response to Referee #3's previous comments. Their comments are below. The reviewers find that the paper has improved in revision, and therefore we'll be happy in principle to publish it in Nature Ecology & Evolution, pending minor revisions to satisfy the reviewers' final requests and to comply with our editorial and formatting guidelines.

[REDACTED]

Reviewer #1 (Remarks to the Author):

Dear editor,

I am satisfied with the response and adjustments made by the authors.

24Reviewer #4 (Remarks to the Author):

In this paper, the authors impute hydraulic traits of woody plant species, quantify community-level hydraulic trait distributions, and use those community-level trait distributions to predict drought-induced mortality events. This is a timely and important topic, the analysis is well reasoned and appropriate, and the results and conclusions are compelling and well presented. I was asked in this review to focus on the authors' responses to reviewer 3's comments, and I feel that they have addressed all of that reviewer's concerns. I have only a few comments of my own.

1) The main goal of the study is to predict drought-induced mortality, and the authors clearly show that their method that includes community-level hydraulic traits outperforms models that include environmental information only. The imputed trait values used to calculate community-level metrics contain environmental information, since 1) hydraulic trait values are strongly influenced by the environment (particularly P_{min} and therefore HSM), and 2) environmental variables were used in the trait imputation. Trait-mortality relationships could therefore reflect direct causal effects of the traits themselves, environmental effects that are indirectly captured by the imputed traits, or a combination. The use of phylogenetic information further obscures the causal link between hydraulic traits and mortality, since the imputed trait values could also carry the signals of other phylogenetically conserved traits. If the goal is pure prediction, this is not a problem, but I think the authors need to be clear that 1) the imputed trait values contain information beyond hydraulic traits, 2) this information may contribute to the improvement in predictive performance, and 3) it is not possible to infer causal effects of hydraulic traits on mortality based on this analysis.

2) The authors briefly mention the issue of sampling bias, but I feel this needs more serious consideration. There could be severe sampling bias in the mortality event data due to variation in sampling effort across space and taxa. This could create spurious trait-mortality relationships, e.g., if environmental factors influence both mortality and sampling effort or if hydraulic traits are correlated with the likelihood of mortality events being observed across species. I feel that the authors should offer a stronger justification of why they think that sampling bias is not a major issue and offer a more thoughtful discussion of how it could influence the results and how it could be addressed in future studies, beyond the statement that "our results will need further confirmation in future works."

Finally, one of the main conclusions is that community-level hydraulic metrics improve predictions of drought-improved mortality compared to species-level mean values, but this is not a fair comparison. Community-level metrics are used to predict community-level mortality, whereas species-level metrics are used to predict species-level mortality, which is likely more difficult. I suggest removing this comparison.

Minor comments

L39: It is not clear what "wall-to-wall" means in this context. Consider rewording.

L289: Include brief description or citation for calculation of "alpha-hull terrestrial range distributions".

Fig. 2, 4: The panels are too small to clearly see what is going on, especially in the maps, which are arguably the most exciting part of the paper. I recommend splitting this into multiple figures or including fewer panels to increase the size of the maps.

Our ref: NATECOLEVOL-220716967B

28th June 2023

Dear Dr. Sanchez-Martinez,

Thank you for your patience as we've prepared the guidelines for final submission of your Nature Ecology & Evolution manuscript, "Increased hydraulic risk in assemblages of woody plant species predicts spatial patterns of drought-induced mortality" (NATECOLEVOL-220716967B). Please carefully follow the step-by-step instructions provided in the attached file, and add a response in each row of the table to indicate the changes that you have made. Please also check and comment on any additional marked-up edits we have proposed within the text. Ensuring that each point is addressed will help to ensure that your revised manuscript can be swiftly handed over to our production team.

****We would like to start working on your revised paper, with all of the requested files and forms, as soon as possible (preferably within two weeks). Please get in contact with us immediately if you anticipate it taking more than two weeks to submit these revised files.****

In recognition of the time and expertise our reviewers provide to Nature Ecology & Evolution's editorial process, we would like to formally acknowledge their contribution to the external peer review of your manuscript entitled "Increased hydraulic risk in assemblages of woody plant species predicts spatial patterns of drought-induced mortality". For those reviewers who give their assent, we will be publishing their names alongside the published article.

Nature Ecology & Evolution offers a Transparent Peer Review option for new original research manuscripts submitted after December 1st, 2019. As part of this initiative, we encourage our authors to support increased transparency into the peer review process by agreeing to have the reviewer

26comments, author rebuttal letters, and editorial decision letters published as a Supplementary item. When you submit your final files please clearly state in your cover letter whether or not you would like to participate in this initiative. Please note that failure to state your preference will result in delays in accepting your manuscript for publication.

Cover suggestions

As you prepare your final files we encourage you to consider whether you have any images or illustrations that may be appropriate for use on the cover of Nature Ecology & Evolution.

Nature Ecology & Evolution has now transitioned to a unified Rights Collection system which will allow our Author Services team to quickly and easily collect the rights and permissions required to publish your work. Approximately 10 days after your paper is formally accepted, you will receive an email in providing you with a link to complete the grant of rights. If your paper is eligible for Open Access, our Author Services team will also be in touch regarding any additional information that may be required to arrange payment for your article.

Please note that *Nature Ecology & Evolution* is a Transformative Journal (TJ). Authors may publish their research with us through the traditional subscription access route or make their paper immediately open access through payment of an article-processing charge (APC). Authors will not be required to make a final decision about access to their article until it has been accepted. [Find out more about Transformative Journals](https://www.springernature.com/gp/open-research/transformative-journals)

Authors may need to take specific actions to achieve [compliance with funder and institutional open access mandates](https://www.springernature.com/gp/open-research/funding/policy-compliance-faqs). If your research is supported by a funder that requires immediate open access (e.g. according to [Plan S principles](https://www.springernature.com/gp/open-research/plan-s-compliance)) then you should select the gold OA route, and we will direct you to the compliant route where possible. For authors selecting the subscription publication route, the journal's standard licensing

27terms will need to be accepted, including <https://www.nature.com/nature-portfolio/editorial-policies/self-archiving-and-license-to-publish>. Those licensing terms will supersede any other terms that the author or any third party may assert apply to any version of the manuscript.

For information regarding our different publishing models please see our <https://www.springernature.com/gp/open-research/transformative-journals> Transformativ Journals page. If you have any questions about costs, Open Access requirements, or our legal forms, please contact ASJournals@springernature.com.

[REDACTED]

[REDACTED]

Reviewer #1:
Remarks to the Author:
Dear editor,

I am satisfied with the response and adjustments made by the authors.

Reviewer #4:
Remarks to the Author:

In this paper, the authors impute hydraulic traits of woody plant species, quantify community-level hydraulic trait distributions, and use those community-level trait distributions to predict drought-induced mortality events. This is a timely and important topic, the analysis is well reasoned and appropriate, and the results and conclusions are compelling and well presented. I was asked in this review to focus on the authors' responses to reviewer 3's comments, and I feel that they have addressed all of that reviewer's concerns. I have only a few comments of my own.

1) The main goal of the study is to predict drought-induced mortality, and the authors clearly show that their method that includes community-level hydraulic traits outperforms models that include environmental information only. The imputed trait values used to calculate community-level metrics contain environmental information, since 1) hydraulic trait values are strongly influenced by the environment (particularly P_{min} and therefore HSM), and 2) environmental variables were used in the

28trait imputation. Trait-mortality relationships could therefore reflect direct causal effects of the traits themselves, environmental effects that are indirectly captured by the imputed traits, or a combination. The use of phylogenetic information further obscures the causal link between hydraulic traits and mortality, since the imputed trait values could also carry the signals of other phylogenetically conserved traits. If the goal is pure prediction, this is not a problem, but I think the authors need to be clear that 1) the imputed trait values contain information beyond hydraulic traits, 2) this information may contribute to the improvement in predictive performance, and 3) it is not possible to infer causal effects of hydraulic traits on mortality based on this analysis.

2) The authors briefly mention the issue of sampling bias, but I feel this needs more serious consideration. There could be severe sampling bias in the mortality event data due to variation in sampling effort across space and taxa. This could create spurious trait-mortality relationships, e.g., if environmental factors influence both mortality and sampling effort or if hydraulic traits are correlated with the likelihood of mortality events being observed across species. I feel that the authors should offer a stronger justification of why they think that sampling bias is not a major issue and offer a more thoughtful discussion of how it could influence the results and how it could be addressed in future studies, beyond the statement that “our results will need further confirmation in future works.”

Finally, one of the main conclusions is that community-level hydraulic metrics improve predictions of drought-improved mortality compared to species-level mean values, but this is not a fair comparison. Community-level metrics are used to predict community-level mortality, whereas species-level metrics are used to predict species-level mortality, which is likely more difficult. I suggest removing this comparison.

Minor comments

L39: It is not clear what “wall-to-wall” means in this context. Consider rewording.

L289: Include brief description or citation for calculation of “alpha-hull terrestrial range distributions”.

Fig. 2, 4: The panels are too small to clearly see what is going on, especially in the maps, which are arguably the most exciting part of the paper. I recommend splitting this into multiple figures or including fewer panels to increase the size of the maps.

Author Rebuttal, second revision:

Comments from the reviewers:

Reviewer #4 (Remarks to the Author):

1) The main goal of the study is to predict drought-induced mortality, and the authors clearly show that their method that includes community-level hydraulic traits outperforms models that include environmental information only. The imputed trait values used to calculate community-level metrics contain environmental information, since 1) hydraulic trait values are strongly influenced by the environment (particularly P_{min} and therefore HSM), and 2) environmental variables were used in the trait imputation. Trait-mortality relationships could therefore reflect direct causal effects of the traits themselves, environmental effects that are indirectly captured by the imputed traits, or a combination. The use of phylogenetic information further obscures the causal link between hydraulic traits and mortality, since the imputed trait values could also carry the signals of other phylogenetically conserved traits. If the goal is pure prediction, this is not a problem, but I think the authors need to be clear that 1) the imputed trait values contain information beyond hydraulic traits, 2) this information may contribute to the improvement in predictive performance, and 3) it is not possible to infer causal effects of hydraulic traits on mortality based on this analysis.

We thank the reviewer for this comment. We completely agree. In the revised version, we explicitly indicate that our framework does not allow for detecting causality (L289-292, tracked changes version). Instead, we show a correlative pattern (association) that likely involves a mechanism that has been detected to have a causal link with stress and mortality in previous studies at finer scales (Anderegg et al., 2015). In the current version of the manuscript, we

explicitly acknowledge that our results using imputed hydraulic traits may result from an environmental signal. In fact, this signal may not be easy to remove from the trait effects, as these traits are tightly related to environmental conditions, especially at a global scale (Sanchez-Martinez et al., 2020). However, note that in our analyses, we include an aridity index as a covariate and hydraulic risk effects remain significant. In this version, we also acknowledge that the use of the phylogeny may include the signal of other important functional traits that present a high phylogenetic signal, which may also be influencing mortality (L291-294, tracked changes version). Moreover, in the reviewed version of this manuscript, we refer to edapho-climatic variables instead of environmental variables for a higher specificity and to avoid misunderstandings.

2) The authors briefly mention the issue of sampling bias, but I feel this needs more serious consideration. There could be severe sampling bias in the mortality event data due to variation in sampling effort across space and taxa. This could create spurious trait-mortality relationships, e.g., if environmental factors influence both mortality and sampling effort or if hydraulic traits are correlated with the likelihood of mortality events being observed across species. I feel that the authors should offer a stronger justification of why they think that sampling bias is not a major issue and offer a more thoughtful discussion of how it could influence the results and how it could be addressed in future studies, beyond the statement that “our results will need further confirmation in future works.”

We thank the reviewer for this thoughtful comment. We agree that sampling bias is an important potential issue in global studies based on available data. Our justification in this regard is based on the fact that the relationship between hydraulic risk and mortality is conserved across biomes. The relationship is not only dependant on data coming from highly sampled biomes such as the Mediterranean or temperate forest, but appears also in generally underrepresented biomes such as tropical forests and boreal forests. In the current version of the manuscript, we include this argument (L271-274, tracked changes version). Moreover, we repeat our methods aggregating mortality occurrences at different scales to minimize spatial autocorrelation, and our results stay consistent. However, we still acknowledge in the text that there may be a sampling bias influencing our results (e.g., 162-164, 274-279, tracked changes version).

Finally, one of the main conclusions is that community-level hydraulic metrics improve predictions of drought-improved mortality compared to species-level mean values, but this is not a fair comparison. Community-level metrics are used to predict community-level mortality, whereas species-level metrics are used to predict species-level mortality, which is likely more difficult. I suggest removing this comparison.

We thank the reviewer for raising this issue and we completely agree with his considerations. We have removed this comparison as an explicit objective in the current version of the manuscript (L105-108, 302-303, tracked changes version), although we maintain the corresponding results, as we fell they provide relevant context to interpret our community-level discussion.

Minor comments

L39: It is not clear what “wall-to-wall” means in this context. Consider rewording.

We agree with the reviewer. We have reworded the sentence removing the “wall-to-wall” expression (L43, tracked changes version).

L289: Include brief description or citation for calculation of “alpha-hull terrestrial range distributions”.

We thank the reviewer for this comment. A reference has been provided in the current version (L316, tracked changes version).

Fig. 2, 4: The panels are too small to clearly see what is going on, especially in the maps, which are arguably the most exciting part of the paper. I recommend splitting this into multiple figures or including fewer panels to increase the size of the maps.

We thank the reviewer for this comment. We have separated plots in figure 2 into 3 figures (Figure 2, 3 and 4) to guide the reader through the results and to allow for higher panel sizes. We have also redistributed the plots in figure 6 so now map panels are bigger.

Final Decision Letter:

26th July 2023

Dear Mr Sanchez-Martinez,

33I am writing in the temporary absence of my colleague Simon Harold to let you know that we are pleased that your Article entitled "Increased hydraulic risk in assemblages of woody plant species predicts spatial patterns of drought-induced mortality", has now been accepted for publication in Nature Ecology & Evolution.

Over the next few weeks, your paper will be copyedited to ensure that it conforms to Nature Ecology and Evolution style. Once your paper is typeset, you will receive an email with a link to choose the appropriate publishing options for your paper and our Author Services team will be in touch regarding any additional information that may be required

Due to the importance of these deadlines, we ask you please us know now whether you will be difficult to contact over the next month. If this is the case, we ask you provide us with the contact information (email, phone and fax) of someone who will be able to check the proofs on your behalf, and who will be available to address any last-minute problems . Once your paper has been scheduled for online publication, the Nature press office will be in touch to confirm the details.

Acceptance of your manuscript is conditional on all authors' agreement with our publication policies (see www.nature.com/authors/policies/index.html). In particular your manuscript must not be published elsewhere and there must be no announcement of the work to any media outlet until the publication date (the day on which it is uploaded onto our web site).

Please note that Nature Ecology & Evolution is a Transformative Journal (TJ). Authors may publish their research with us through the traditional subscription access route or make their paper immediately open access through payment of an article-processing charge (APC). Authors will not be required to make a final decision about access to their article until it has been accepted. Find out more about Transformative Journals

Authors may need to take specific actions to achieve compliance with funder and institutional open access mandates. If your research is supported by a funder that requires immediate open access (e.g. according to Plan S principles) then you should select the gold OA route, and we will direct you to the compliant route where possible. For authors selecting the subscription publication route, the journal's standard licensing terms will need to be accepted, including https://www.nature.com/nature-portfolio/editorial-policies/self-archiving-and-license-to-publish. Those licensing terms will supersede any other terms that the author or any third party may assert apply to any version of the manuscript.

If you have any questions about our publishing options, costs, Open Access requirements, or our legal

forms, please contact ASJournals@springernature.com

We welcome the submission of potential cover material (including a short caption of around 40 words) related to your manuscript; suggestions should be sent to Nature Ecology & Evolution as electronic files (the image should be 300 dpi at 210 x 297 mm in either TIFF or JPEG format). Please note that such pictures should be selected more for their aesthetic appeal than for their scientific content, and that colour images work better than black and white or grayscale images. Please do not try to design a cover with the Nature Ecology & Evolution logo etc., and please do not submit composites of images related to your work. I am sure you will understand that we cannot make any promise as to whether any of your suggestions might be selected for the cover of the journal.

You can generate the link yourself when you receive your article DOI by entering it here: <http://authors.springernature.com/share>.

[REDACTED]

P.S. Click on the following link if you would like to recommend Nature Ecology & Evolution to your librarian <http://www.nature.com/subscriptions/recommend.html#forms>

** Visit the Springer Nature Editorial and Publishing website at http://editorial-jobs.springernature.com?utm_source=ejp_NEcoE_email&utm_medium=ejp_NEcoE_email&utm_campaign=ejp_NEcoE for more information about our career opportunities. If you have any questions please click [here](mailto:editorial.publishing.jobs@springernature.com).**